# The C9ORF72 GGGGCC expansion forms RNA G-quadruplex inclusions and sequesters hnRNP H to disrupt splicing in ALS brains

Erin G Conlon[1], Lei Lu[2], Aarti Sharma[2], Takashi Yamazaki[1], Timothy Tang[1], Neil A Shneider[2]*, James L Manley[1]*

[1]Department of Biological Sciences, Columbia University, New York, United States; [2]Department of Neurology, Columbia University Medical Center, New York, United States

**Abstract** An expanded GGGGCC hexanucleotide in *C9ORF72* (C9) is the most frequent known cause of amyotrophic lateral sclerosis (ALS) and frontotemporal dementia (FTD). It has been proposed that expanded transcripts adopt G-quadruplex (G-Q) structures and associate with proteins, but whether this occurs and contributes to disease is unknown. Here we show first that the protein that predominantly associates with GGGGCC repeat RNA in vitro is the splicing factor hnRNP H, and that this interaction is linked to G-Q formation. We then show that G-Q RNA foci are more abundant in C9 ALS patient fibroblasts and astrocytes compared to those without the expansion, and more frequently colocalize with hnRNP H. Importantly, we demonstrate dysregulated splicing of multiple known hnRNP H-target transcripts in C9 patient brains, which correlates with elevated insoluble hnRNP H/G-Q aggregates. Together, our data implicate C9 expansion-mediated sequestration of hnRNP H as a significant contributor to neurodegeneration in C9 ALS/FTD.

*For correspondence: ns327@columbia.edu (NAS); jlm2@columbia.edu (JLM)

## Introduction

Amyotrophic lateral sclerosis (ALS) and frontotemporal dementia (FTD) are two related syndromes that occur alone or together in families, and sporadically in individuals. Expansion of a hexanucleotide, GGGGCC ($G_4C_2$) in an intervening sequence separating two putative first, noncoding exons in the gene *C9ORF72* (C9) (*DeJesus-Hernandez et al., 2011*; *Renton et al., 2011*) is the most frequent known cause of both disorders. How this expansion leads to disease is unclear, although several non-mutually exclusive mechanisms have been suggested. The existence of ribonucleoprotein inclusions involving the RNA binding proteins TDP-43 or FUS in the brains and spinal cords of nearly all ALS patients and the occurrence of ALS-causing mutations in their respective genes (*Ling et al., 2013*) suggest that defects in RNA processing lead to neurodegeneration. In the case of C9ALS, sequestration of RNA-binding proteins by the transcribed $G_4C_2$ expansion is a proposed pathogenic mechanism, and it has been shown that RNA from this locus forms intranuclear foci in the brains of affected individuals (*Donnelly et al., 2013*). However, while a number of proteins have been suggested to bind the transcribed repeats (*Haeusler et al., 2014*; *Lee et al., 2013*; *Cooper-Knock et al., 2014*), the functional significance of such binding is not well established. It has also been shown that poly dipeptide repeat (DPR) proteins are translated from repeat-containing transcripts, and DPRs may contribute toxic effects of their own (*Mori et al., 2013b*; *Kwon et al., 2014*; *Wen et al., 2014*).

While the mechanism(s) of C9-mediated neurodegeneration is (are) unknown, two properties of the pathogenic mutation are noteworthy: First, expansion length is variable, with expansions above an indeterminate threshold (~30–40, but up to several thousand) causing disease (*Nordin et al., 2015*). Second, the sequence of the expansion favors formation of an especially stable G quadruplex (G-Q) structure, which consists of planar tetrad arrays of four non-sequential guanosine residues connected by Hoogsteen hydrogen bonds. Two or more of these planar arrays then interact through pi-pi stacking, stabilized by a central monovalent cation, generally potassium (*Davis, 2004*). Independent G-Qs can stack upon each other, forming higher order multimers. This is relevant to sequences that have many repetitive G-Q motifs in a series, such as is in telomeres and in expanded regions like C9 (*Patel et al., 2007*; *Martadinata et al., 2011*; *Payet and Huppert, 2012*; *Kobayashi et al., 2011*).

Several factors influence G-Q stability. For example, stability is greater for RNA versus equivalent DNA structures, reflecting the presence of 2′ hydroxyls that participate in H-bonds within the RNA quadruplex (*Collie et al., 2011*). Furthermore, four quartet G-Qs, such as the *O. nova* telomeric sequence $(GGGGTTTT)_n$, are substantially more stable than their three quartet human telomeric counterparts, which in turn are more stable than G-Qs with two quartets (*Lee et al., 2008*; *Mullen et al., 2012*). The fact that RNA G-Qs, and more so those with four quartets such as the C9 expansion, are so exceptionally stable is important when considering toxic function arising from repeat RNA in ALS/FTD. It has been proposed that G-Q formation is a significant aspect of the toxicity of the repeats in ALS/FTD (*Haeusler et al., 2014*; *Simone et al., 2015*), but whether such structures are indeed prevalent in C9 ALS/FTD patient brains, and if so, whether, and how, they are pathogenic in ALS/FTD has not been determined.

Here we provide insights into the molecular behavior of multimeric RNA G-Qs encoded by C9 repeats, which suggest how they may contribute to neurodegeneration in ALS/FTD. Using a sensitive UV-crosslinking assay, we identified the splicing factor hnRNP H as the predominant C9 RNA binding protein in a brain cell-derived nuclear extract, and show how this interaction reflects association with G-Q containing repeats. Significantly, using G-Q- and hnRNP H-specific antibodies for immunofluorescence, we observed more G-Q structures and increased hnRNP H colocalization in C9 patient-derived cells than in control cells. We detected aberrant alternative splicing of multiple known hnRNP H splicing targets in C9 patient brains, and show that this correlated with increased levels of insoluble G-Q associated hnRNP H in these same brain samples. Intriguingly, the magnitude of splicing deregulation we observed in these specimens also roughly correlated with disease severity. Our results thus support a model in which the $G_4C_2$ repeat expansions in C9 RNA form stable G-Q structures that sequester hnRNP H in insoluble aggregates, which in turn lead to changes in hnRNP H-target exon inclusion in a variety of disease-relevant transcripts.

## Results

### Characterization of C9 protein binding

We first wished to identify protein(s) that bind C9 repeat RNA, and to investigate the effects of RNA structure on binding. While previous studies have identified a number of possible C9 RNA binding proteins (*Haeusler et al., 2014*; *Lee et al., 2013*; *Cooper-Knock et al., 2014*; see Discussion), we investigated this in a way that allowed us not only to analyze direct binding with sensitivity and selectivity, but also to determine the possible influence of G-Q formation on binding. To this end, we incubated 32P CTP-labeled in vitro transcribed RNAs containing ten C9 repeats (10R), typically at a concentration of ~1 nM, in nuclear extract prepared from U87 glioblastoma cells (NE) and exposed the samples to UV light. Crosslinked RNA-protein complexes were digested with RNAse A, T1, a mixture of both, or nothing, and resolved by SDS-PAGE. We consistently observed two predominant bands, one of ~60–65 kDa and one ~75 kDa (*Figure 1a*). Unexpectedly, these species were unaffected by RNAse A, T1 or both (*Figure 1a*, lanes 2–5), with the only difference being that in the absence of RNAse, a band appeared at the top of the gel, possibly reflecting higher-order aggregates that failed to migrate into the gel (lane 2). Notably, although the RNA alone ran as a single band of the predicted size on denaturing urea PAGE (*Figure 1b*, lanes 1 and 5, bottom), on SDS PAGE the free RNA gave rise to two bands, one at the expected MW of the full-length RNA (~20–25 kDa, based on protein markers), and a lower band that migrated faster (~15 kDa) (*Figure 1a*,

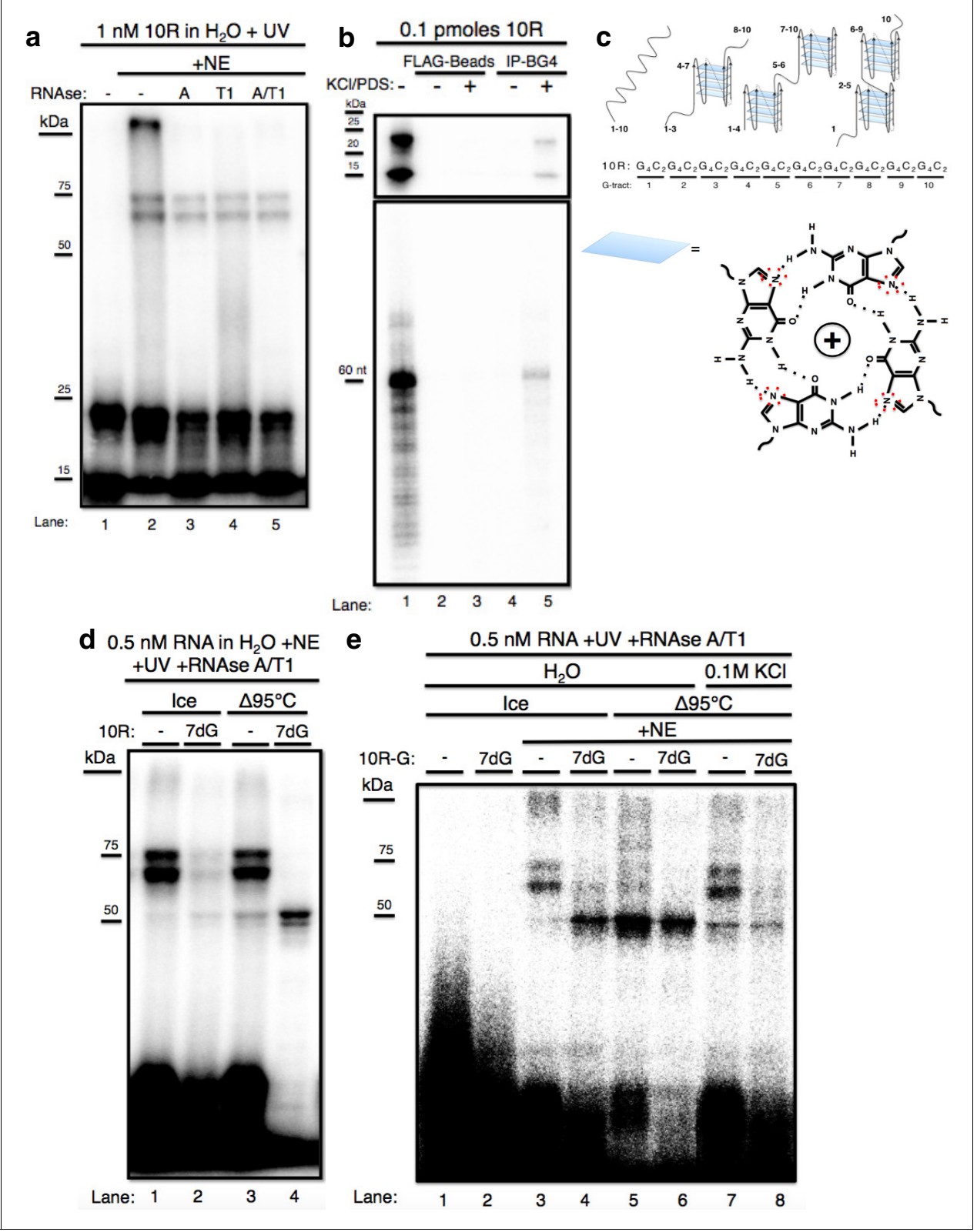

**Figure 1.** The GGGGCC expansion forms stable multimeric G-quadruplexes (G-Qs). (a) 10R UV crosslinked to U87 nuclear extract (NE) and digested with RNAse A (0.5 ug) (lane 3), RNAse T1 (10U) (lane 4), both (0.5ug RNAse A/10U RNAse T1) (lane 5), or nothing (lane 2), and separated by 10% SDS-PAGE. (b) 0.1 pmoles of 10R in water (lanes 1, 2 and 4) or 200 mM KCl and 500 µM pyridostatin (PDS) (lanes 3 and 5), IPed with BG4 antibody (lanes 4 and 5) or beads alone (lanes 2 and 3). Products were separated by 10% SDS-PAGE (top) and by 8M Urea/10% PAGE (bottom). (c) Representative G-Q

*Figure 1 continued on next page*

*Figure 1 continued*

folding conformations of 10 repeats of GGGGCC RNA. Numbers indicate G-tracts that participate in GQs or linear fragments. The right most conformation represents 8 consecutive G-tracts forming an octamer. Beneath, a depiction of one quartet, with the N7 (replaced with C in 7-deaza GTP) position surrounded by red dashed oval (d) 10R (lanes 1 and 3) and 10R-7dG (lanes 2 and 4) heated to 95℃ (lanes 3 and 4) or left on ice (lanes 1 and 2), crosslinked to NE, digested with RNAse A/T1 (.5 ug/10U) and separated by 10% SDS-PAGE. (e) 10R-G (lanes 1, 3, 5 and 7) and 10R-G7dG (lanes 2, 4, 6 and 8) heated to 95℃ in water (lanes 5 and 6) or 100 mM KCl (lanes 7 and 8) or left on ice (lanes 3 and 4), crosslinked to NE, digested with RNAse A/T1 (0.5 µg/10 U) and separated by 10% SDS-PAGE. This gel was cropped to remove two intervening lanes between lanes 6 and 7.

The following figure supplement is available for figure 1:

**Figure supplement 1.** Immunoprecipitation of 10R with BG4.

lane 1). The fact that RNAse had no effect on the mobility of the crosslinked doublet suggested that the shifted RNA was in fact still full-length and thus resistant to RNAse. Given that the free RNA also resolved into two bands, we hypothesized that the two higher MW species resulted from crosslinking of the same protein to two differentially mobile full-length RNAs.

We next wished to investigate why free RNA resolved into two bands on SDS PAGE, and whether these species might explain why the crosslinked RNA also appeared as two bands. It has been reported that transcripts containing multiple independent G-Qs can give rise to distinct species on native gels due to multiple folded conformations (*Payet and Huppert, 2012*). To test whether the RNAs we detected represent different G-Q conformations, we immunoprecipitated (IPed)10R in either water or G-Q stabilizing conditions of 200 mM KCl and 500 µM pyridostatin (PDS) using the G-Q specific antibody BG4 (*Biffi et al., 2013*), and resolved the IPed RNA by nondenaturing (*Figure 1b*, top) or denaturing (*Figure 1b*, bottom) PAGE. On denaturing PAGE, full-length RNA was detected following IP, consistent with the presence of G-Q (lane 5, bottom). On nondenaturing PAGE, both products were recovered following IP (lane 5, top), indicating that both species contain G-Qs. Pre-incubation of the RNA in KCl and PDS enhanced the efficiency of IP (lane 5 versus lane 4); however, G-Q stabilization was not necessary for recognition of 10R by BG4, as we observed minor recovery of 10R in water, visible upon longer exposure (*Figure 1—figure supplement 1*, lane 4). We conclude that the two 10R species resolved by nondenaturing PAGE reflect two distinct G-Q states, e.g., one in which four $G_4C_2$ tracts form one G-Q, and the other in which eight tracts form two G-Qs (*Figure 1c*).

We next wished to determine whether the crosslinked doublet was in fact due to two differentially mobile G-Q structures bound to one protein. To this end, we attempted to denature RNA structures by heating in water to 95 ℃ either 10R or a derivative (10R-7dG) in which GTP was replaced during transcription with the G-Q destabilizing GTP analog 7-deaza GTP (7dG). RNAs were then added to NE, crosslinked, treated with RNAses and the pattern following SDS PAGE compared to RNA that had been kept on ice prior to crosslinking (*Figure 1d*). Heating 10R slightly reduced the appearance of the crosslinked doublet, leading to a corresponding increase of a band at ~50 kDa (compare lanes 1 and 3). More striking results were observed with 10R-7dG. Even in the absence of heating, crosslinking to produce the high MW doublet was substantially reduced relative to 10R (lane 2), supporting the view that these species represent protein binding to G-Q RNA. Strikingly, heating 10R-7dG abolished appearance of the doublet and produced strong crosslinking that gave rise to the ~50 kDa species (lane 4). (A weaker band was detected just below the 50 kDa species; as described below, the two are very likely related). Notably, the only free RNA (at the bottom of the gel) that underwent significant RNAse degradation was heated 10R-7dG (lane 4), consistent with the idea that RNAse resistance reflects stable G-Q structures.

It is conceivable that the identity of the labeled nucleotide could influence the pattern of crosslinking, and using a different label could thus provide additional insights into the interactions described above. We therefore repeated the above experiments with $^{32}$P-GTP instead of $^{32}$P-CTP in the transcription reaction, generating 10R-G and 10R-G-7dG, and used these RNAs for crosslinking as above (*Figure 1e*). Unheated 10R-G reproduced the doublet pattern observed with 10R (lane 3). However, heated 10R-G resulted in the nearly complete loss of the doublet and appearance of increased amounts of the 50 kDa species (and fainter band beneath), as well as a reduction in intensity of the upper free RNA species (lane 5). With unheated 10R-G-7dG we observed much higher

levels of 50 kDa bands than with 10R (compare lanes 2 and 4). When 10R-G-7dG was heated, only the 50 kDa bands remained, and the free RNA underwent significant degradation, similar to what was observed with 10R-7dG (compare lanes 4 and 6). To extend these results, we incubated 10R-G and 10R-G7dG in 0.1 M KCl before heating, reasoning that stabilization of G-Qs might counteract denaturation. Indeed, crosslinking to 10R-G incubated in 0.1 M KCl led to less crosslinking at 50 kDa and reappearance of the higher MW doublet (compare lane 7 to lane 5). Likewise, heating 10R-G7dG in 0.1 M KCl reduced appearance of the 50 kDa band, while also resulting in a faint higher MW doublet (lane 8). These results both strengthen our hypothesis that the crosslinked species all arise from interaction with the same protein, and also indicate that the identity of the labeled nucleotide can indeed influence crosslinking and/or protection from RNAse.

## HnRNP H is the major crosslinked protein

We next wished to identify the crosslinking protein(s) described above. Several lines of evidence, including the size of the 'denatured' crosslinked species (~50 kDa) and known affinity for poly(rG), led us to hypothesize that it may be the splicing regulator hnRNP H (*Caputi and Zahler, 2002*; *Chou et al., 1999*; *Dardenne et al., 2014*; *Fisette et al., 2012*). To determine whether the crosslinked band(s) observed above represent(s) RNA bound to hnRNP H, we employed crosslinking followed by IP with anti-hnRNP H antibodies. With 10R-G and 10R-G7dG (*Figure 2a*) we once again detected crosslinked species at 50 kDa (lanes 2 and 6, respectively), both of which were IPed by hnRNP H antibodies (lanes 4 and 8, respectively). With 10R-G, and to a lesser extent 10R-G7dG, we again observed that a significant portion of the crosslinked complex failed to migrate into the gel (lanes 2 and 6). Significantly, these species were also recovered with hnRNP H antibody (lanes 4 and 8) but not with beads alone (lanes 3 and7). Some crosslinked doublet was visible with 10R-G (lane 2), although no doublet was IPed (lane 4; see below). We also performed crosslinking and IP with the $^{32}$P-CTP-labeled RNA (*Figure 2b*). With 10R-7dG, the major and minor 50 kDa crosslinked species described above were IPed, but the more abundant higher MW species again were not (lane 8). With 10R, the higher MW doublet was again the major species detected, and was also not IPed (lanes 2–4). Nonetheless, we believe this doublet does in fact represent hnRNP H, but the structured nature of the RNA that enables the complex to resist RNAse degradation (*Figure 1a*) also obstructs recognition by hnRNP H antibody (see also below). Note that the band just beneath the 50 kDa species may represent the highly related protein hnRNP H2, which would be predicted to IP with the hnRNP H antibody (*Stark et al., 2011*).

The above data provided evidence that the 50 kDa species arose from crosslinking of hnRNP H and related proteins, but were inconclusive regarding the higher MW doublet. Although the data are consistent with the doublet also arising from hnRNP H crosslinking, we utilized two additional approaches to investigate this further. First, we purified hnRNP H1- 6XHIS from *E. coli* (H1; *Figure 2c*) and performed UV-crosslinking assays with H1 incubated with 10R-G that was digested with RNAse A/T1 or not, and compared the results with the pattern in NE (*Figure 2d*). In this case, to enhance the potential G-Q formation and thus detection of the doublet, we included 50 mM KCl in the buffer. Crosslinking with NE followed by RNAse treatment produced a combination of the doublet and ~50 kDa species (lane 3). Notably, the amount of doublet was much greater, and the amount of the 50 kDa species much less, than when crosslinking was performed in the absence of KCl (compare with *Figure 1e*, lane 5). Crosslinking with purified H1 followed by RNAse treatment led to a single species running slightly above 50 kDa (lane 1), reflecting the ~4 kDa affinity tag on H1. However, when products were not treated with RNAse, both H1 and NE produced highly similar patterns of doublets, as well as lower mobility species that failed to migrate into the gel, likely reflecting larger multimers as observed above (*Figure 2b*, lane 3). Notably, when we heated the RNA even more stringently (to 100°C) and omitted KCl (*Figure 2e*), we observed a much simpler pattern, essentially showing only the 50 kDa (NE, lane 3) or 54 kDa (H1, lane 4) species.

As a second approach to determine whether the crosslinked doublet reflects hnRNP H crosslinking, we asked whether siRNA-mediated depletion of hnRNP H reduces formation of these products. To this end, we prepared two U87 cell NEs, one treated with an siRNA targeting hnRNP H (siH) and the other with a negative control siRNA (siNC). We were unable to deplete HnRNP H completely, but reduced its levels in siH NE to roughly 45% relative to siNC NE (*Figure 2F*, top right; silver stained gel (lower right) confirmed equivalent protein concentrations in two NEs). We then performed UV crosslinking with 10R-G in both NEs. Importantly, both species in the crosslinked

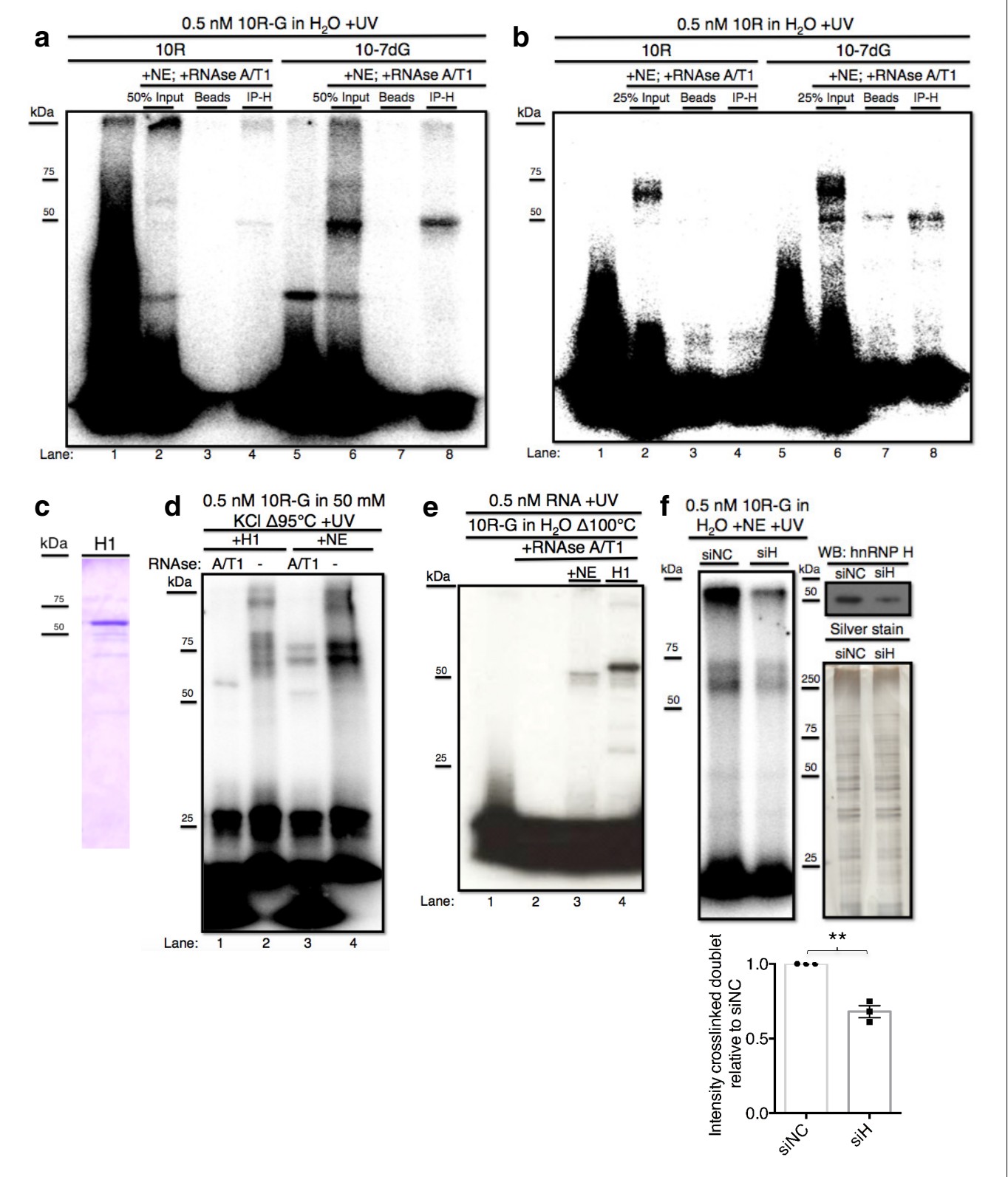

**Figure 2.** hnRNP H associates with ten G4C2 repeat RNA in vitro. (**a**) IP of 10R-G (lane 5) and 10R-7dG (lane 10) crosslinked to U87 nuclear extract (NE), digested with RNAse A/T1 (0.5 ug/10U) and separated by 10% SDS-PAGE. (**b**) IP of 10R (lane 4) and 10R-G7dG (lane 8) crosslinked to U87 NE, digested with RNAse A/T1 (.5 ug/10U) and separated by 10% SDS-PAGE. (**c**) 6X-HIS tagged hnRNP H1 (H1) produced in *E. coli*. (**d**) 10R-G heated to 95°C in 50 mM KCl and crosslinked to purified H1 (lanes 1 and 2) or NE (lanes 3 and 4) and digested with RNAse A/T1 (.5 ug/10U) (lanes 1 and 3) or nothing (lanes

*Figure 2 continued on next page*

*Figure 2 continued*

2 and 4), and separated by 10% SDS-PAGE. (e) 10R-G heated to 100°C in $H_2O$ and crosslinked to NE (lane 3) or purified H1 (4) and digested with RNAse A/T1 (0.5 ug/10U) (lanes 3 and 4). (f) (Left panel)10R-G crosslinked to NE prepared from U87 cells treated with negative control siRNA (siNC) or siRNA against hnRNP H (siH), digested with RNAse A/T1 (0.5 μg/10 U) and separated by 10% SDS-PAGE. (Right panel, upper) Western blot of hnRNP H in siNC and siH nuclear extracts. (Right panel, lower) Silver stain of siNC and siH nuclear extracts. (Lower panel) Quantification of crosslinked doublet from three experiments with siNC and siH nuclear extracts.

The following figure supplement is available for figure 2:

**Figure supplement 1.** Gel shift assays with 4R and 4R-7dG.

doublet, as well as the band at the top of the gel, were significantly reduced in siH NE relative to the amount detected in siNC NE (*Figure 2F*). We conclude that hnRNP H is the predominant, and perhaps only, protein in U87 NE to crosslink to 10R and that the variations in mobility reflect differential G-Q formation and/or stability.

We next examined hnRNP H binding to C9 RNA by another method, gel shift assays. This approach allowed us to estimate the affinity of hnRNP H for C9 RNA, as well as examine any quantitative preference for the folded versus unfolded conformation. For this, we used RNAs with four $G_4C_2$ repeats, as the ten repeat RNAs produced complex patterns in these assays. RNAs were transcribed with GTP or 7-deaza-GTP to generate 4R and 4R-7dG and gel shift assays were performed with increasing concentrations of purified recombinant H1 (*Figure 2—figure supplement 1*). From these assays, we estimated the $K_D$ for the interaction between H1 and 4R to be 75.5 ± 14 nM, while the affinity of H1 for 4R-7dG was significantly higher, with a $K_D$ of 13.5±3.0 nM. Notably, we observed the appearance of supershifted RNA-protein complexes with both RNAs at higher concentrations of H1. This may reflect multiple H1 molecules binding to a single RNA, and is perhaps related to the higher-order structures observed in UV-crosslinking experiments. These results suggest that hnRNP H binds either to 4R RNA in both folded and unfolded conformations but with different modes of recognition and affinities, or preferentially to linear G-tracts, as has been found for the third qRRM domain of the highly related protein hnRNP F (*Dominguez et al., 2010*; *Samatanga et al., 2013*). Either mechanism, however, has relevance to the interaction of hnRNP H with the C9 repeats, as expanded transcripts are predicted to be composites of G-Qs and linear G-tracts. While determining the precise mode of binding will likely require structural studies, these results provide evidence that hnRNP H binds the C9 repeats with sufficient affinity for the interaction to be physiologically significant.

## Excessive G-Q structure accumulate in C9 patient derived cells

We next wished to determine the possible relevance of C9 G-Q formation to ALS. Previous efforts to visualize GGGGCC repeat foci in cells have relied on in situ hybridization (*Donnelly et al., 2013*; *Cooper-Knock et al., 2014*). However, this approach on the one hand may underestimate the extent of repeat foci because of the difficulties denaturing putative G-Q structures, while on the other provides no information on the structure of the repeats. We therefore asked instead whether an excess of G-Q structures exists in cells from ALS patients with the C9 expansion (C9ALS). We used the (Flag-tagged) BG4 antibody to visualize G-Qs by immunofluorescence of fibroblasts and astrocytes from C9ALS patients and from healthy individuals without neurological symptoms (all patient-derived samples described in *Supplementary file 1*).

To quantitate BG4 foci, we developed the program 'BG4 Count' in Matlab (see Materials and methods). The program displays only the BG4 foci (red) that coincide with DAPI staining in patient and control fibroblasts (*Figure 3a*) and astrocytes (*Figure 3b*). We chose to limit our analysis of BG4 to DAPI-positive areas in order to avoid counting artifacts, since the majority of G-Q staining occurs in the nucleus and because DAPI staining in the cytoplasm can indicate presence of nucleic acids. Omitting BG4 but not FLAG antibody resulted in no staining in the red channel (*Figure 3—figure supplement 1*). Due to potential variability in staining between experiments, we analyzed each experiment separately, normalizing the average counts per cell in each cell line to the average control cell in each experiment, providing a fold change relative to control. From four separate staining experiments with fibroblasts, we compared 16 normalized values from seven C9ALS patients

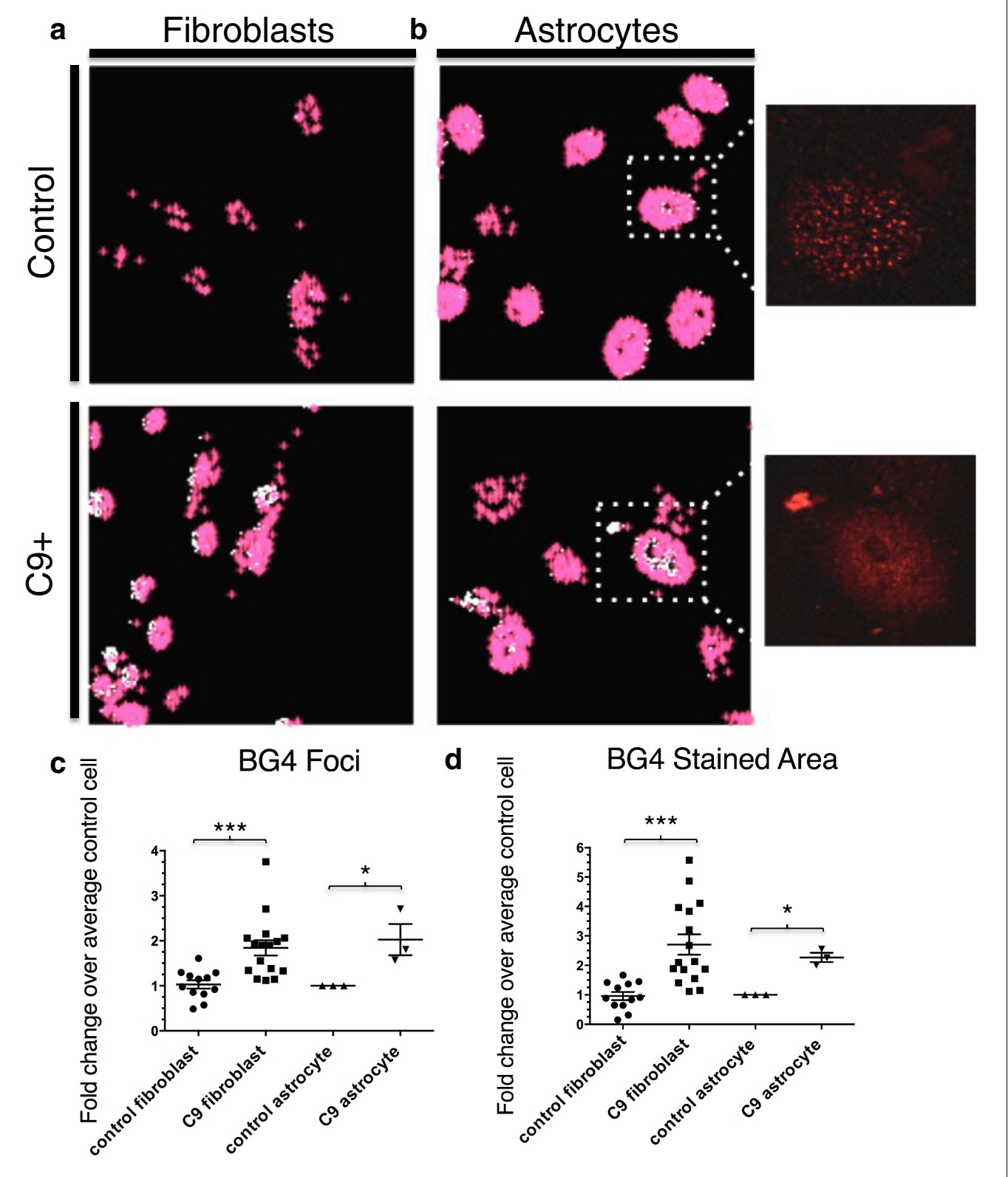

**Figure 3.** Quantification of BG4 stained foci and area in fibroblasts and astrocytes from C9 ALS/FTD patients and controls. Representative nonALS and C9ALS fibroblasts (a) and astrocytes (b) showing the projection created by 'BG4 Count' that represents all stained area above the determined threshold (0.1) in red, with areas of particularly dense staining in white. Inset depicts source image showing only the red (BG4-FLAG) channel. (c) Number of foci and total stained area (d) as fold change over the average number of foci, or area, respectively, per control cell for C9 fibroblasts (n replicates = 16), *Figure 3 continued on next page*

*Figure 3 continued*

control fibroblasts (n replicates = 12), C9 astrocytes (n replicates = 3), and control astrocytes (n replicates = 3) (*p<0.05, **p<0.01, ***p<0.001). Error bars are plotted to SEM.

The following figure supplements are available for figure 3:

**Figure supplement 1.** Immunofluorescence with FLAG antibody without prior BG4 incubation.

**Figure supplement 2.** BG4 quantification of RNAse A treated astrocytes.

comprising a total of ~1860 cells to 12 values from five control lines comprising a total of 1350 cells using an unpaired t-test, with all C9ALS fibroblasts comprising one group, and all control fibroblasts another. This indicated that, on average, patient cells contained 1.8 times as many G-Q foci as control cells, with a false discovery rate of less than 0.1% (*Figure 3c*). Likewise, comparison of one C9ALS patient astrocyte line and one control over three separate experiments (210 and 227 total cells, respectively) revealed that patient cells had 2.0 times as many foci, with a false discovery rate of less than 5% (*Figure 3c*).

We also compared the total area of foci in patient and control cells. Comparison of the average stained areas across the sets of cells through an unpaired t-test revealed that patient fibroblasts contained on average 2.8 times as much stained area per cell than did control fibroblasts, with a false discovery rate of less than 0.1%, while patient astrocytes had 2.3 times as much area, with a false discovery rate of less than 1.0% (*Figure 3d*). We conclude from these experiments that BG4 foci are larger and more numerous in cells from C9ALS patients than from healthy controls.

We next wished to provide evidence that at least a fraction of the BG4 foci detected were RNA and not DNA. BG4 can recognize both DNA and RNA G-Qs, and both have been detected in cells (*Biffi et al., 2013*, *2014*). Treatment of astrocytes with RNAse A prior to BG4 antibody incubation in fact reduced the number of BG4 foci per cell, by 27% in the control patient cells and by a striking 76% in the C9ALS cells (relative to untreated control and C9 cells, respectively; *Figure 3—figure supplement 2*). Given that the untreated C9ALS astrocytes in this experiment had 2.7 times as many foci per cell as untreated control cells, RNAse A treatment thus reduced both patient and control to nearly equivalent numbers of foci per cell (112 and 125 foci per cell, respectively), indicating that the excess of BG4 foci observed in the patients is in fact attributable to RNA G-Qs.

## hnRNP H colocalizes with G-Q aggregates in C9 patient derived cells, but not in control cells

We next investigated whether G-Q foci colocalize with hnRNP H and if so, whether this occurs more frequently in C9ALS patient cells than in controls. To address this, we first analyzed the same sets of fibroblasts and astrocytes by coimmunofluorescence in the nucleus (*Figure 4a and b*) or DAPI-positive cytoplasm (*Figure 4c and d*; see below). We counted each of the individual colocalization events, and then divided these values by the total number of cells (210 C9ALS and 227 control). The approximate incidence of nuclear and cytoplasmic colocalizing foci detected in C9ALS astrocytes was 6.2 and 8.1%, respectively, and 0.4 and 2.6% in the control cells (*Figure 4e*). The one nuclear colocalization event we detected in the control astrocytes was much smaller and less distinct than any of the events detected in the C9ALS astrocytes (*Figure 4f*). In fibroblasts, we counted 17 instances of high colocalization in the nuclei of C9ALS fibroblasts (*Figure 4g*), and 24 in the cytoplasm (*Figure 4h*), or 0.9% and 1.3%, respectively, for the ~1860 cells we counted. No nuclear colocalization events were observed in any of the ~1350 control fibroblasts, while cytoplasmic colocalization was observed in ~0.2% of cells (*Figure 4i*).

Although the incidence of colocalization events detected in the above experiments was small, likely reflecting technical difficulties in visualizing the overlapping foci, the differences between patient and control cells were highly significant. One potential explanation for the low frequency of overlapping G-Q/hnRNP H foci we observed is that an excess of BG4-reactive RNA masked hnRNP H within RNA-bound aggregates. In support of this, when we counted colocalization events after RNAse A treatment in 44 patient astrocytes we found roughly the same rate of cytoplasmic

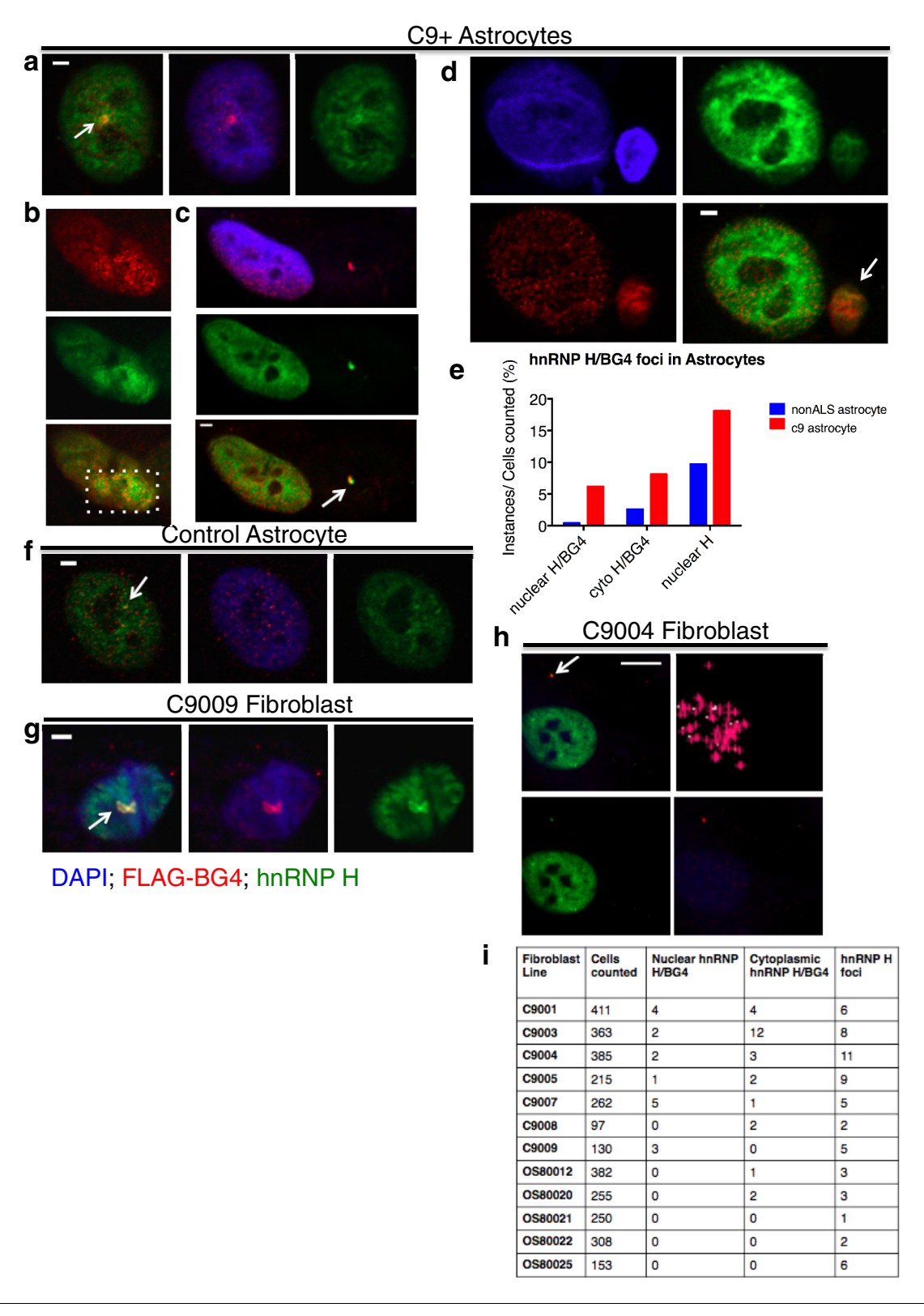

DAPI; FLAG-BG4; hnRNP H

| Fibroblast Line | Cells counted | Nuclear hnRNP H/BG4 | Cytoplasmic hnRNP H/BG4 | hnRNP H foci |
|---|---|---|---|---|
| C9001 | 411 | 4 | 4 | 6 |
| C9003 | 363 | 2 | 12 | 8 |
| C9004 | 385 | 2 | 3 | 11 |
| C9005 | 215 | 1 | 2 | 9 |
| C9007 | 262 | 5 | 1 | 5 |
| C9008 | 97 | 0 | 2 | 2 |
| C9009 | 130 | 3 | 0 | 5 |
| OS80012 | 382 | 0 | 1 | 3 |
| OS80020 | 255 | 0 | 2 | 3 |
| OS80021 | 250 | 0 | 0 | 1 |
| OS80022 | 308 | 0 | 0 | 2 |
| OS80025 | 153 | 0 | 0 | 6 |

**Figure 4.** Visualization of nuclear and cytoplasmic BG4/hnRNP H colocalization in C9 patient cells and controls. C9 patient astrocytes with nuclear hnRNP H/BG4 foci (a, c) nuclear redistribution of hnRNP H towards high BG4 staining (b) and large cytoplasmic foci (d). (e) Percentage of control and C9 patient astrocytes containing nuclear and cytoplasmic hnRNP H colocalization events. (f) The single, small nuclear hnRNP H/BG4 nuclear foci in control astrocyte. (g) Nuclear hnRNP H/BG4 inclusion in C9 patient fibroblasts. (h) C9 patient fibroblasts with cytoplasmic hnRNP H/BG4 foci in DAPI-
*Figure 4 continued on next page*

*Figure 4 continued*

positive cytoplasm. (i) Summary of all fibroblasts counted and the number of colocalization event. All objects are indicated with arrow or dashed box (c), and all scale bars = 3 microns with the exception of (h), 10 microns.

The following figure supplements are available for figure 4:

**Figure supplement 1.** hnRNP H/BG4 colocalization in events in RNAse A treated astrocytes.

**Figure supplement 2.** hnRNP H foci in post-mortem human spinal motor neurons.

colocalization events (9% after treatment as compared to 8.1% before), but more than three times as many nuclear colocalization events (20% after treatment compared to 6.2% before) (*Figure 4— figure supplement 1*). No colocalization events, nuclear or cytoplasmic, were detected in control astrocytes after RNAse A treatment. We conclude that C9ALS cells contain hnRNP H/G-Q foci that are virtually absent in control non-ALS cells.

A related possibility is that some hnRNP H/G-Q foci were refractive to BG4 staining. We therefore also counted hnRNP H foci not overlapping with BG4 in the fibroblasts and astrocytes analyzed above, and again found that the C9ALS cells had more foci than the controls. Specifically, C9ALS astrocytes had nonoverlapping hnRNP H foci in 18% of cells and control astrocytes had foci in 10% of cells (*Figure 4e*), while C9ALS fibroblasts had hnRNP H foci in 2.5% of cells and 1.1% of control fibroblasts had foci (*Figure 4i*).

We next wished to extend these results to a more physiological context, ALS patient brains. For this, we used post-mortem spinal cord sections from two C9ALS patients and two nonALS controls (*Supplementary file 1*) for immunostaining with hnRNP H antibodies. (BG4 staining was not possible due to high variability and background.) We counted instances of nuclear hnRNP H foci that were larger than 0.5 microns in any dimension, which was approximately the lower limit for measurement of discernable puncta. Motor neurons (MNs) were identified based on their location in the ventral horn, size, and low DAPI signal. In the 36 C9ALS MNs we detected, we counted a total of 46 large hnRNP H foci, representing possible C9 RNA/hnRNP aggregates (see below). In contrast, in 19 non-ALS MNs, we detected only three large hnRNP H foci. We conclude that hnRNP H distribution is significantly altered in disease-relevant MN populations in C9ALS patients.

## C9ALS patient brains show missplicing of hnRNP H target transcripts relative to healthy brains

We next investigated whether the G-Q/hnRNP H association has functional consequences in ALS patient brains. We hypothesized that binding of hnRNP H to the C9 repeats leads to local depletion of hnRNP H, and the consequent deregulation of splicing targets. Such a mechanism would be similar to that which occurs in myotonic dystrophy type 1 and 2 (*Wheeler and Thornton, 2007*). In the case of the C9 repeats, sequestration of hnRNP H would most frequently result in increased skipping of cassette exons whose inclusion is mediated by concentration-dependent binding of hnRNP H to G-runs enriched in the immediate downstream intron (*Xiao et al., 2009*). To investigate this possibility, we first searched RNA-seq datasets from hnRNP H knockdown (KD) and CLIP experiments (*Xiao et al., 2009*) to identify potential hnRNP H targets of interest to the ALS/FTD disease continuum. We selected 13 exons (listed in *Supplementary file 2*) whose inclusion appeared significantly reduced upon hnRNP H KD (*Xiao et al., 2009*). Targets were selected on the basis of direct links to ALS/FTD or other neurological diseases (e.g. *ATXN2, KIF1C, PEX19, ARRB2*), or because of documented involvement in pathways implicated in ALS/FTD such as excitotoxictiy (*GABBR1, CABIN1*), protein degradation (*PSMD4, BAT3*), cytoskeletal remodeling and neuritogenesis (*PPP1R12C*) and RNA metabolism (*PAN2, RNH1, DAZAP1*). For comparison, we also analyzed three cassette exons, in the genes *WDR45, MELK* and *PFKM*, that are regulated by the splicing factor SRSF2 (*Zhang et al., 2015a*) but whose response to hnRNP H KD had not previously been studied.

We analyzed alternative splicing events in C9ALS and non-ALS control cerebellum. We used cerebellum because it undergoes significant atrophy across the ALS/FTD disease spectrum (*Tan et al., 2014*), and also because of tissue availability and RNA quality. We purified total RNA from

postmortem cerebellum of fourteen individuals, six without neurological symptoms (ages 54–89, avg. age = 67 years), one sporadic SOD1 ALS case (age 52), and seven C9ALS patients (ages 58–73, avg. age = 63 years) (*Supplementary file 1*). Due to the fundamental pathological, and likely mechanistic, differences between SOD1- and C9-ALS we grouped this SOD1 patient with the nonALS patients. For comparison, we also prepared RNA from U87 cells that were treated with siRNA against hnRNP H or a negative control siRNA (*Figure 5a*). We performed $^{32}$P-RT-PCR for all samples, amplifying across alternatively spliced cassette exons (diagrammed *Figure 5b*) and quantitated the proportion of exon-inclusion products. We then determined a mean percent inclusion for each patient or KD condition based on three or four independent repeats (*Figure 5c–l*, *Figure 5—figure supplement 1*), grouped each patient as control or C9ALS and compared the values using an unpaired t-test, with significance set at $p < 0.05$.

We first verified splicing changes in the siRNA-treated U87 cells. We found decreased exon inclusion following KD for 12/15 hnRNP H target transcripts (*Figure 5d–l*; *Figure 5—figure supplement 1b–d*), while three were unchanged (*Figure 5—figure supplement 1a,e and f*). Unexpectedly, two of the three SRSF2 target exons, in *WDR45* (*Figure 5c*) and *PFKM* (*Figure 5—figure supplement 1f*), showed increased inclusion upon hnRNP H KD. Although neither of these exons appeared among those detected by hnRNP H-CLIP-seq (*Xiao et al., 2009*), the skipped exon in *WDR45* contains two potential G-Q motifs that could serve as hnRNP H binding sites, while the *PFKM* exon also contains several G-runs. In contrast to the activating effect observed when hnRNP H binds intronic elements, hnRNP H generally represses inclusion when it binds exonic sites (*Figure 5b*) (*Xiao et al., 2009*), consistent with our results.

We next compared the inclusion values of the 15 hnRNP H and three SRSF2 target exons between control non-ALS and C9ALS samples. Strikingly, 11 of the 12 hnRNP H targets and one of the two SRSF2 targets that changed following KD in U87 cells displayed similar changes in the C9ALS samples compared to controls (*Figure 5c–l*, *Figure 5—figure supplement 1b and c*; note that for two exons, in *ARRB2* and *KIAA1542*, p values were 0.065 and 0.081, respectively). The hnRNP H target exon in *CABIN1* and the SRSF2 target in *PFKM* displayed no difference between controls and C9 patients (*Figure 5—figure supplement 1d and g*), nor did two of the three hnRNP H target exons that were unaffected following KD (*Figure 5—figure supplement 1e and f*). However, the target exon in *DAZAP1* that did not change following hnRNP H KD was significantly more excluded in C9ALS relative to control (*Figure 5—figure supplement 1a*). The unaffected SRSF2 target transcript, *MELK,* was not detected in brain tissue (*Figure 5—figure supplement 1h*).

Widespread splicing dysregulation has been previously reported in the brains of C9ALS patients, with motif analysis suggesting hnRNP H as a potential driver of the reported changes (*Prudencio et al., 2015*). To extend our analysis, we examined RNA-seq from patient brains (*Prudencio et al., 2015*) and hnRNP H KD (*Xiao et al., 2009*) to identify events common to the two datasets. We performed $^{32}$P-RT-PCR on nine cassette exon targets that were found significantly altered in both datasets. For two genes, *UQCRB* and *UBE2I*, we could detect only exon exclusion (*Supplementary file 2*), while another, *TCEB1*, varied greatly in the overall expression across our cohort of patients, and could not be accurately quantified (data not shown). For the other six targets we confirmed that the regulated exon was more included in patients than controls in transcripts from *HNRPDL, OS9* and *RPL10* (*Figure 5—figure supplement 2a–d*) and more excluded in patients than controls in *RBM38, PCBP2* and *SIGMAR1* transcripts (*Figure 5—figure supplement 2d–f*), all entirely consistent with the results of the hnRNP H KD RNA-seq (*Supplementary file 2*).

We next investigated whether the affected exons we have described are indeed direct targets of hnRNP H. To address this, we examined previous hnRNP H CLIP-seq data (*Xiao et al., 2009*; *Katz et al., 2010*) to determine whether hnRNP H physically binds in the vicinity of the regulated exons (*Figure 5—figure supplement 4*). Indeed, we found CLIP tags within 200 bps of 17 out of 18 exons that differed between nonALS and C9ALS, with the only exception being the exon in *ATXN2*, for which CLIP tags were present within 500 bp. These observations strongly support a direct role of hnRNP H in these misregulated splicing events.

In addition to the direct involvement of hnRNP H, it is possible that secondary effects may contribute to the splicing changes detected in C9ALS. For example, changes in expression of splicing factors with hnRNP H-regulated exons, such as RBM38, PCBP2 and HNRPDL, could alter splicing. Similarly, levels of the splicing factor hnRNP A1, which has been suggested to bind the C9 repeats (*Cooper-Knock et al., 2014*; *Zamiri et al., 2014*), have been suggested to influence roughly one

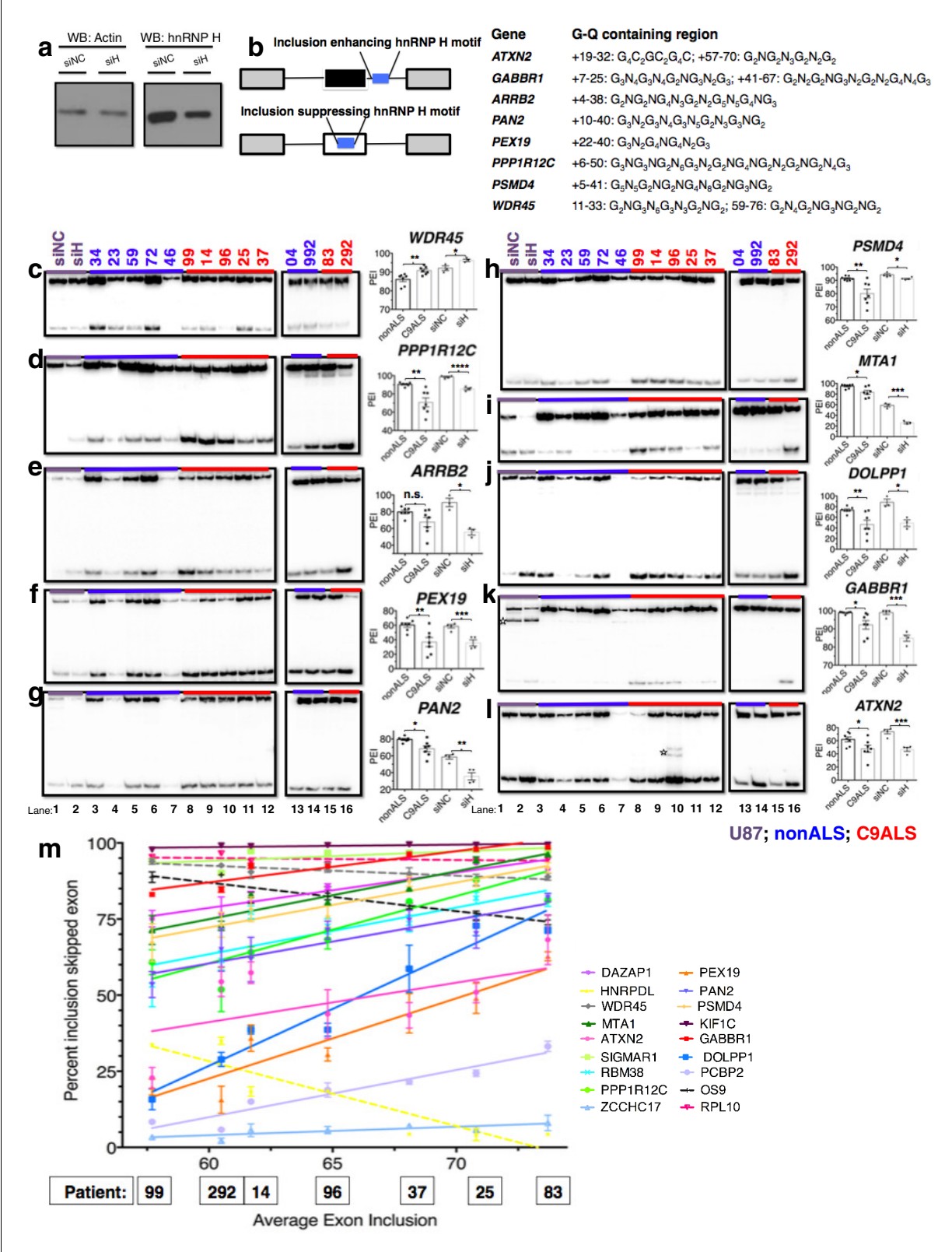

**Figure 5.** Missplicing of hnRNP H target exons in C9 ALS patient brains. (**a**) Western blot of hnRNP H knockdown in U87 cells. (**b**) (Left) Diagram of representative hnRNP H target exons with small blue rectangles depicting the relative locations of inclusion-enhancing intronic (upper) and repressive exonic (lower) G-Q binding motifs, listed at right. (**c–l**) Representative hnRNP H target exons perturbed in C9ALS patient brains. (Left) $^{32}$P-RT-PCR across alternatively spliced exon for siRNA-treated U87 cells (lanes 1 and 2), nonALS control cerebellum (lanes 3–7, 13 and 14) and C9ALS cerebellum

*Figure 5 continued on next page*

*Figure 5 continued*

(lanes 8–12, 15 and 16). Star symbols (**k** and **l**) denote non-specific PCR artifact bands. (Right) Graph depicting the average percent inclusion of each target exon, all values listed in *Supplementary file 2*. PEI = percent exon inclusion. For lanes 4 and 7, see sample quality note in *Supplementary file 1* (*p<0.05, **p<0.01, ***p<0.001, ****p<0.0001). Error bars are plotted to SEM. (**m**) Linear regression of percent inclusion values for each of the 18 significantly affected (p<0.05) target exons as a function of the average percent inclusion of these genes in each C9ALS sample. Solid lines are used for genes with positive slopes, dashed lines for genes with negative slopes. Error bars are plotted to SEM.

The following figure supplements are available for figure 5:

**Figure supplement 1.** (Left) $^{32}$P-RT-PCR across alternatively spliced exons for siRNA-treated U87 cells (**a–h** lanes 1 and 2), nonALS control cerebellum (**a,b**) lanes 3–9; **c–f** lanes 3–7, 13 and 14; **g,h** lanes 3–7) and C9ALS cerebellum (**a,b** 10–16; **c–f** lanes 8–12, 15 and 16; **g,h** lanes 8–12).

**Figure supplement 2.** (Left) $^{32}$P-RT-PCR across alternatively spliced exons for nonALS control cerebellum (lanes 1–7) and C9ALS cerebellum (lanes 8–14).

**Figure supplement 3.** (Left) $^{32}$P-RT-PCR across alternatively spliced exons for nonALS control cerebellum (lanes 1–7) and C9ALS cerebellum (lanes 8–14).

**Figure supplement 4.** CLIP-seq of hnRNP A1 and representative hnRNP H target exons.

third of the events that change upon hnRNP H depletion (*Huelga et al., 2012*). We therefore asked whether cassette exons that are regulated by hnRNP A1 are misspliced in C9ALS relative to nonALS brains. We performed $^{32}$P-RT-PCR across six exons affected by hnRNP A1 KD: three that undergo more skipping following hnRNP A1 depletion and three that are more included (*Huelga et al., 2012*) (*Figure 5—figure supplement 3d–f*). Only one exon, in *ZCCHC17*, differed significantly, and it was more excluded in C9ALS patients relative to nonALS, opposite the outcome that would be predicted if hnRNP A1 levels were reduced. Notably, the regulated exon in *ZCCHC17* had hnRNP H CLIP tags, ~100 bp downstream, likely coinciding with a potentially G-Q forming intronic sequence (*Figure 5—figure supplement 4*). Only one other gene, *G2AD (AP1G2)*, had CLIP tags in or near the regulated exon, overlapping a polyG exonic motif ~70 bp downstream (*Figure 5—figure supplement 4*). This exon was indeed included less in C9ALS patients (p=0.096), again opposite the effect predicted for hnRNP A1 depletion (*Supplementary file 2*).

We next asked whether each C9ALS patient displayed a similar extent of exon exclusion for all of the affected transcripts. To investigate this, we determined the mean percent exon inclusion of the 16 significantly misspliced hnRNP H target exons and the single SRSF2 and hnRNP A1 targets for each patient, and ordered the patients from lowest to highest average inclusion. We reasoned that if there was variation, the patient in which hnRNP H function was most reduced would have the lowest inclusion values, while the patient with the least functional reduction would have the highest exon inclusion values, for all of the genes, except for *WDR45, HNRPDL, RPL10* and *OS9*, which we hypothesized would display the opposite trend. Using linear regression, we plotted a best-fit line for each gene (*Figure 5m*), and indeed found that for every gene except *WDR45, HNRPDL, RPL10* and *OS9* a straight line with a positive slope could be drawn (values listed in *Supplementary file 2*), demonstrating that the patient with the lowest average inclusion had the lowest percent inclusion for all genes tested. As expected, the lines for *WDR45, HNRPDL, RPL10* and *OS9* all had negative slopes. Interestingly, this trend also reflected disease duration, i.e., time from onset of symptoms until death, for each patient, a rough indicator of disease severity (*Supplementary file 1*). Shorter disease duration correlated with greater changes in exon inclusion, suggesting that hnRNP H splicing dysregulation is a marker of clinical progression.

## hnRNP H forms insoluble aggregates with G-Qs in C9 patient brains

We next investigated whether the G-Q/hnRNP H foci we detected by immunostaining of patient cells sequester sufficient amounts of hnRNP H to bring about the splicing changes we detected in C9ALS brains. To address this, we analyzed motor cortex (Brodmann Area 4) from the seven C9ALS patients and seven controls in our cohort by biochemical fractionation with Sarkosyl-containing buffers, a method used previously to detect protein aggregates from the brains of Alzheimer's patients

(*Ren and Sahara, 2013*), followed initially by western blotting for hnRNP H (*Figure 6a*). We quantified the abundance of hnRNP H in each fraction, summing them to find the amount of total hnRNP H, and determined the percentage of insoluble hnRNP H from two independent experiments. We plotted these values for C9ALS patients and nonALS controls and compared them using a t-test (*Figure 6b*). The average percentage of insoluble hnRNP H was 30.0% in nonALS and 56.1% in C9ALS (p=0.013). We conclude that there is 1.9 times as much insoluble hnRNP H in the motor

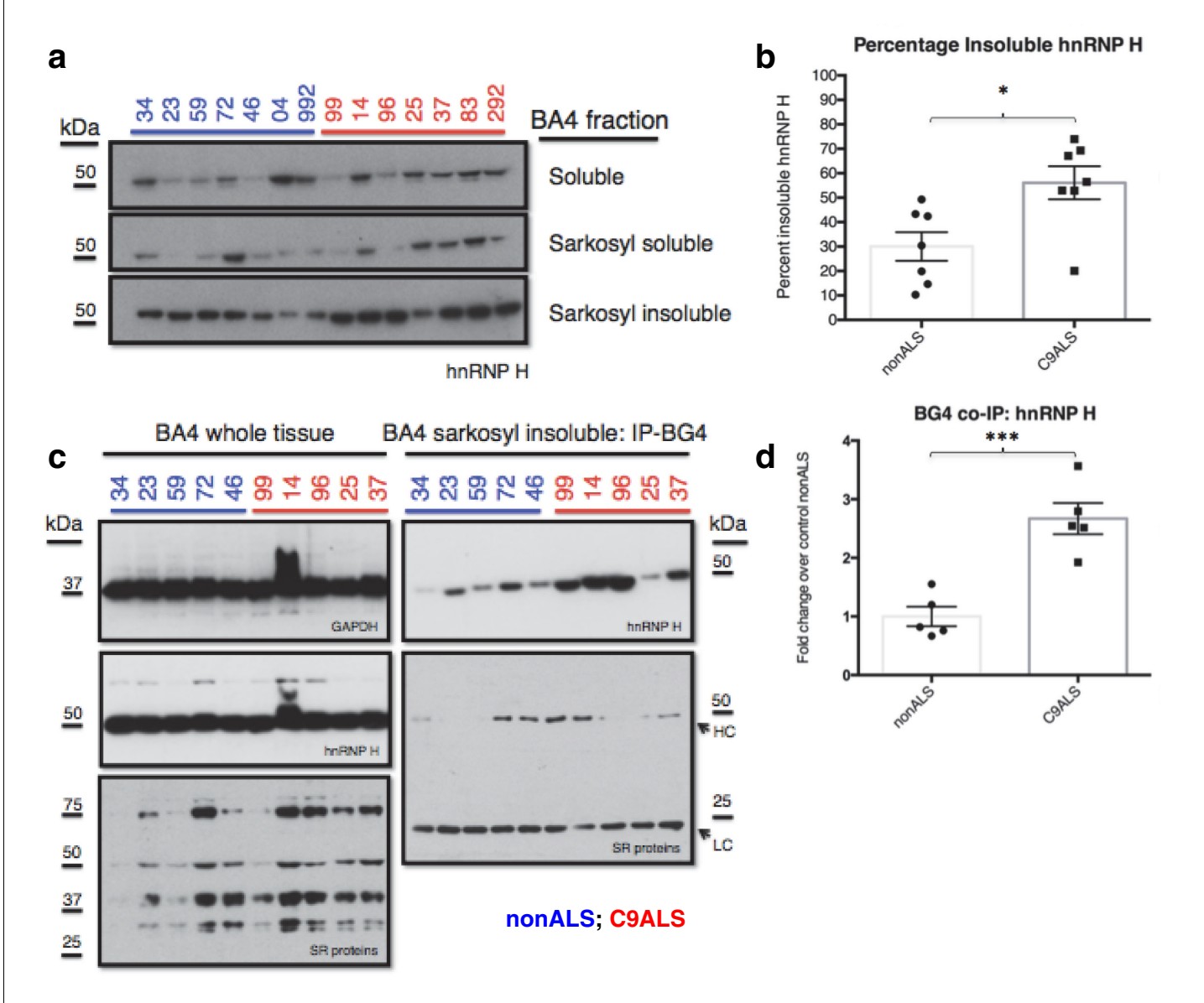

**Figure 6.** C9ALS brains are enriched with insoluble G-Q bound hnRNP H. (a) Western blot analysis of hnRNP H in each of three fractions (soluble, sarkosyl soluble, sarkosyl insoluble). (b) Graph of the percentage of insoluble hnRNP H in each of the 14 patients (c) IP with BG4 (IP-BG4) from the resuspended sarkosyl insoluble fraction of motor cortex (BA4). (Left) Western blots of input unfractionated tissue probed with anti-GAPDH, hnRNP H and SR protein antibodies. (Right) Western blot of IP-BG4 probed with anti-hnRNP H and SR protein antibodies. HC and LC refer to IgG heavy chain and light chain, respectively. (d) Quantification of hnRNP H co-IPed with BG4 (n = 3 replicates) in each of ten non-ALS and C9ALS samples. (*p<0.05, **p<0.01, ***p<0.001, ****p<0.0001). Error bars are plotted to SEM.

The following figure supplement is available for figure 6:

**Figure supplement 1.** Co-immunoprecipitation of BG4 from the Sarkosyl insoluble fraction of patient motor cortex (BA4).

cortex of C9ALS patients as compared to control, providing evidence that hnRNP H sequestration by C9 repeats occurs in the brains of ALS patients to a biochemically significant level.

We hypothesized that the change in distribution of hnRNP H reflects the formation of insoluble aggregates with C9 G-Q RNA. To provide evidence for this, we performed co-IP with BG4 and the resuspended insoluble pellets from motor cortex from five C9ALS patients and five controls, followed by western blotting for hnRNP H, and for comparison, for SR proteins, including SRSF1 and SRSF2, which reportedly bind the C9 repeats (*Figure 6c*). We performed three independent co-IP experiments, and quantified the amount of hnRNP H that co-IPed with BG4 in C9ALS and non-ALS controls. We found, on average, 2.7 times as much hnRNP H co-IPed from the C9ALS samples as from the nonALS controls (*Figure 6d*) (p=0.0007). We obtained similar results when we performed no BG4, beads-alone controls and subtracted the hnRNP H signals without antibody from the signal with BG4 for each sample: 2.8 times as much hnRNP H co-IPed from the C9ALS samples as from the nonALS controls (*Figure 6—figure supplement 1*). Using an antibody that recognizes a common phospho-epitope on all SR proteins, we did not detect co-IP of any SR proteins (*Figure 6c*). We conclude that hnRNP H forms aggregates with G-Qs in the brains of C9ALS patients, providing additional evidence that hnRNP H is indeed sequestered by C9 G-Q RNA, and to an extent sufficient to bring about the observed splicing dysregulation. Below we discuss how the effects on splicing, and the aggregation itself, may be relevant to the ALS/FTD disease spectrum.

## Discussion

A number of studies over the last few years have identified proteins capable of interacting with the C9 repeats in vitro and which could potentially become sequestered by expanded transcripts in vivo. While a majority of these studies did not consider how G-Q formation might affect protein binding, some took explicit consideration of the G-Q structure formed by the repeats (*Haeusler et al., 2014*; *Reddy et al., 2013*; *Zamiri et al., 2014*), while another rejected G-Q-binding candidates by discarding proteins that exhibited similar binding to scrambled targets that would also form entirely GC G-Qs (*Donnelly et al., 2013*). The list of possible C9 repeat binding proteins currently includes hnRNP A3 (*Mori et al., 2013a*), SRSF1 (*Lee et al., 2013*; *Reddy et al., 2013*; *Zamiri et al., 2014*), Pur alpha (*Rossi et al., 2015*; *Xu et al., 2013*), ADARB2 (*Donnelly et al., 2013*), SRSF2 (*Cooper-Knock et al., 2014*; *Lee et al., 2013*), hnRNP H (*Cooper-Knock et al., 2014*; *Lee et al., 2013*), hnRNP A1 (*Cooper-Knock et al., 2014*; *Zamiri et al., 2014*), ALYREF (*Cooper-Knock et al., 2014*), nucleolin (*Haeusler et al., 2014*) and RanGAP (*Zhang et al., 2015b*). Of these proteins, only SRSF1, hnRNP H and nucleolin have been suggested to bind RNA G-Qs (*von Hacht et al., 2014*); however, the known binding specificities of SRSF1 and nucleolin appear inconsistent with high affinity binding to GGGGCC repeats (*Tacke and Manley, 1995*; *Ghisolfi-Nieto et al., 1996*), and our results did not provide evidence for SR proteins binding C9 RNA. Moreover, many of these studies initially found an excessive number of candidates, in the hundreds, which required further refinement (*Cooper-Knock et al., 2014*; *Donnelly et al., 2013*; *Haeusler et al., 2014*; *Mori et al., 2013a*).

The above studies paint a complex and somewhat confusing picture of GGGGCC repeat recognition by RNA binding proteins. What distinguishes our findings from these earlier ones is that, as assayed by sensitive UV crosslinking, we consistently observed predominant binding of a single protein, which most often appeared as multiple bands due to the folded structure of the ten repeat RNA. Our conclusion that hnRNP H is the major C9 binding protein is consistent with studies implicating it in in C9ALS (*Lee et al., 2013*) as well as in G-Q binding (*Dardenne et al., 2014*; *Fisette et al., 2012*; *von Hacht et al., 2014*). Moreover, a meta-analysis of independent RNA pull-down studies identified hnRNP H1 or the highly related hnRNP H3 as the proteins most consistently found associated with C9 RNA (*Haeusler et al., 2016*). Thusprovides compelling evidence not only for what may be an emerging consensus, but also for the functional significance of this interaction to ALS/FTD.

Our data are also entirely consistent with the known properties of hnRNP H as a splicing regulator. KD of hnRNP H followed by RNA-seq revealed that exons that respond with significantly lowered inclusion show enrichment of G-runs within 70 bases downstream of their 5′ splices site (*Xiao et al., 2009*). Moreover, a positive correlation between the extent of the change in inclusion and the number of G's in these G-runs was observed, and larger changes in splicing following KD were detected for exons with relatively weak 5′ splice sites. Thus, when hnRNP H is depleted, or

sequestered as we propose, these types of exons would be the most severely compromised. Indeed, sequencing of RNA from C9ALS patient brains revealed global changes in alternative splicing, particularly skipping of exons that are proximally enriched for putative hnRNP H binding motifs (*Prudencio et al., 2015*). Our work provides a mechanism to explain these findings, and implicates G-Q formation and hnRNP H sequestration in the pathogenesis of ALS/FTD.

Exon skipping is the most common form of alternative splicing, and hnRNP H levels influence a significant number of these events. While it is unknown which, if any, of the splicing changes we detected are disease relevant, all of the target genes we analyzed encode proteins that can be mechanistically linked to ALS/FTD. For example, *ATXN2* encodes a polyglutamine protein associated with spinocerebellar ataxia 2 (*Imbert et al., 1996*) and contains an expansion that also confers increased susceptibility to ALS in sporadic (*Elden et al., 2010*), and possibly C9 (*van Blitterswijk et al., 2014*), cases. Additionally, *GABBR1* encodes one of the subunits of the heterodimeric GABA receptor B (GABA$_B$R). When exon 19 is skipped, as we observed in patient brains, a frameshift occurs, likely reducing levels of GABBR1 protein. GABA is the major inhibitory neurotransmitter in the mammalian CNS (*Gassmann and Bettler, 2012*) and reduced GABA-mediated inhibition may contribute to excitotoxic cell death in ALS (*Brockington et al., 2013*), which may provide a molecular basis for the potentially pathogenic finding of cortical hyperexcitability in C9ALS patients (*Geevasinga et al., 2015*). Additionally, several of the other hnRNP H-regulated genes we identified, such as *PAN2* (*Udagawa et al., 2015*), *PEX19* (*Murakami et al., 2013*) and *PSMD4* (*Gorrie et al., 2014*), have been indirectly implicated in ALS.

A critical question is whether the hnRNP H sequestration we have described plays a direct, causal role in ALS. While there is no conclusive data establishing causality for any of the multiple mechanisms proposed to explain the toxicity of the C9 repeats, our data and that of others (*Cooper-Knock et al., 2015*) suggest that the extent of splicing dysregulation, in our case specifically of hnRNP H targets, is inversely correlated with disease duration, or directly correlated with disease severity. While analysis of larger cohorts is necessary to confirm this correlation, our findings support a causal role of hnRNP H deficiency in ALS/FTD.

An interesting question is how the C9 repeats, by sequestration of hnRNP H or indeed any proposed mechanism, lead to neurodegeneration in an age-dependent manner. We hypothesize that slight alterations in expression of the genes described above, and perhaps other hnRNP H-regulated transcripts, together contribute to ALS/FTD pathology. We propose that over time, RNA transcribed from the C9 repeats eventually accumulates to levels sufficient to titrate cellular supplies of hnRNP H, thus leading to splicing defects of target transcripts. Some of these changes, such as with genes we have analyzed, would be 'driver' events, others, perhaps most, would be 'passenger' events that do not contribute to disease. While deleterious consequences arising from missplicing might be tolerated or avoided in dividing cells that can regenerate before hnRNP H levels become sufficiently low, longer-lived, postmitotic neurons would not have the advantage of clearing RNA-protein aggregates, which may themselves have toxic consequences for the cell. We predict that in neurons, sequestration of hnRNP H becomes more severe with time and continuing C9 transcription, until functional levels become low enough to deregulate splicing, providing an explanation for the age dependence of ALS/FTD.

Toxic protein aggregates are hallmarks of many neurodegenerative diseases. Within the ALS/FTD spectrum, the composition and location of aggregates are useful for defining distinct disease subtypes that often reflect the diverse genetic underpinnings of these two related disorders (*Thomas et al., 2013*). The most prevalent aggregations in ALS are ubiquitinated, cytoplasmic inclusions containing TDP-43, which occur in the brain and spinal cord of virtually all ALS patients, with the exception of SOD1 and FUS ALS cases. Thus far, the only aggregates unique to C9 are p62/ubiquitin-positive, TDP-43-negative cytoplasmic and nuclear inclusions containing DPRs derived from non-ATG translation of expanded C9 transcripts (*Blokhuis et al., 2013*). Such DPRs can display neurotoxic properties (Mizielinska et al., 2014; *Wen et al., 2014*), consistent with a role in MN death in ALS. However, it has been reported that they are enriched in non-vulnerable brain regions and largely absent from the spinal cord of C9ALS patients, raising questions regarding their exact role in motor neuron degeneration (Davidson et al., 2015). While foci containing expanded C9 RNA have been referred to as aggregates (*Lee et al., 2013*; *Reddy et al., 2013*; *Zamiri et al., 2014*), whether this is in fact the case had not been previously investigated. Our experiments thus provide the first evidence that G-Q expansions bound to hnRNP H indeed form insoluble aggregates, and are

present in disease-relevant regions of the brain. While we have focused on the consequences of hnRNP H sequestration on splicing, it remains to be seen if these aggregates also have dominant neurotoxic properties.

The insights we provide here into the behavior of the C9 G-Qs suggest a possible therapeutic approach that specifically targets the C9 G-Qs and interferes with hnRNP H binding. Many G-Q-interacting compounds exist that have the potential to disrupt hnRNP H/G-Q interactions generically (*Patel et al., 2007*; *Balasubramanian and Neidle, 2009*). However, such compounds would likely also interfere with the normal binding of hnRNP H to intronic G-Qs and thereby exacerbate the splicing defects arising from hnRNP H sequestration. It may be possible however to exploit the high stability, compactness, and symmetry of the C9 G-Q to design drugs that selectively disrupt the pathogenic binding of hnRNP H and thus restore proper hnRNP H stoichiometry and ameliorate disease-relevant splicing changes.

## Materials and methods

### In vitro transcription and probe purification

In vitro transcription reactions were mixed to final concentrations of 1X transcription buffer (supplied by manufacturer, Thermo Scientific, Waltham, MA), 10 mM DTT, 1 U/ul RNAse inhibitor, 500 µM each ATP, CTP, UTP, 20 µM total GTP including 16 pmol $\alpha$-$^{32}$P-GTP (3000 Ci/mmol, Perkin Elmer), 2 µg double stranded T7 promoter containing template (Integrated DNA Technologies, *Supplementary file 3*), and 1 U/µl T7 RNA polymerase (Promega). Reactions were incubated at 37°C for 5 hr. Alternatively, RNA was transcribed cold using the same conditions, except with 500 µM each unlabeled XTP. At the end of the incubation DNAse RQ1 was added to a concentration of one unit/ug of starting DNA template, with DNAse RQ1 buffer (Promega), and incubated for 20 min at 37°C. The reaction mixture was then adjusted to 200 ul, extracted with phenol/chloroform, and the aqueous phase precipitated with $NH_4OAc$, isopropanol and linear polyacrylamide (LPA) (Sigma) for 1 hr-overnight at −20°C. RNA was pelleted in an Eppendorf for 15 min at 21,000 ×g at 4°C, dried, and resuspended in a 1:1 mixture of 50% glycerol and 2X RNA load dye (Ambion). Samples were then boiled and loaded onto an 8M Urea/8% acrylamide gel and electrophoresed at 200V for 2 hr. The gel was transferred to Whatman paper, wrapped in plastic, and exposed to X-ray film for up to 1 hr. Bands corresponding to the desired products were excised and eluted into a solution of 0.75 $NH_4OAc$ and 0.1% SDS for at least 16 hr at 4°C. The elution mixture was then precipitated with approximately 1.5 volumes of cold isopropanol, 1 µl LPA, incubated at −20°C, pelleted as above, dried, and resuspended in 20 µl $H_2O$. Yield was determined by liquid scintillation counts (MicroBeta Trilux) of 1 µl of the final product in $H_2O$ compared to reference of 1 µl of $\alpha$-$^{32}$P-CTP or GTP (Perkin Elmer).

### Purification of BG4 from *E. coli*

The pSang10 vector encoding the BG4 antibody was a generous gift from Dr. Balasubramanian (Cambridge, UK). BG4 antibody was purified by the following protocol: Plasmid was transformed into BL21 Rosetta Cells and a 2 mL culture was grown with 2% glucose, 50 µg/ml Kanamycin, and 35 µg/ml chloramphenicol overnight at 200 rpm and 30°C. This culture was then diluted to 200 mL in 2XYT containing kanamycin and chloramphenicol, and mixed with 100 mL auto-induction media (*Supplementary file 4*), grown at 250 rpm and 37°C for 6 hrs, and then overnight at 280 rpm and 25°C. The culture was pelleted, lysed in TES buffer (50 mM Tris/HCl pH 8.0, 1 mM EDTA pH 8.0, 20% sucrose), purified Ni-NTA Agarose beads (Qiagen) with two washes (PBS + 100 mM NaCl + 10 mM imidazole, pH 8.0), eluted into PBS with 300 mM imidazole, dialyzed against nuclear extract buffer D (*Kataoka and Dreyfuss, 2008*), aliquoted, and stored at −80°C.

### Immunoprecipitation with BG4

FLAG-M2 coated magnetic beads (Sigma) were equilibrated three times with IP buffer (150 mM NaCl, 50 mM Tris pH 7.5, 1 mM EDTA, 10% glycerol, 0.5% NP-40) before rocking with ~5 ug BG4 (described above) or without for 1 hr at 4°C, followed again by three washes with IP buffer. Meanwhile, approximately 0.1 pmoles in vitro transcribed 10R-C was incubated at room temperature for 10 min in either DePC-treated $H_2O$ or 200 mM KCl and 500 µM Pyridostatin (PDS) (Sigma). Equal

amounts of RNA in water and KCl were then incubated with either BG4-caoted beads or beads alone for 1.5 hrs at 4°C, followed by three washes with IP buffer adjusted to 250 mM NaCl. After the final wash, input and beads were incubated with 2X RNA load dye (Ambion) at 95°C for 5 min, rapidly cooled on ice, separated on a magnetic stand, and the supernatant loaded onto a 12% 8 M Urea gel and electrophoresed for 2.5 hr at 400V. For SDS PAGE, samples were prepared identically, except the beads were heated at 95°C for 5 min in 2X SDS PAGE buffer and loaded onto a 10% SDS gel and electrophoresed for 120 min at 100V. Gels were then dried on Whatman paper, wrapped in plastic, and analyzed by phosphorimager (Molecular Dynamics Storm 860).

## UV crosslinking

U87 glioblastoma cells were acquired from previous Manley lab members. The cell line was not authenticated, and was not tested for mycoplasma. Cells were cultured in DMEM with 10% FBS in 140 mm dishes. Nuclear extract (NE) was prepared from confluent cells according to (*Kataoka and Dreyfuss, 2008*). Approximately 1 nM RNA in $H_2O$ was incubated in microcentrifuge tubes with buffer D with KCl adjusted to a final concentration of 100 mM (or otherwise, where indicated) in a total reaction volume of ~10 ul. For denaturation (where applicable), RNA was heated as indicated in a circulating water bath prior to incubation with protein. Four ul of NE was added last, or buffer D in the case of probe alone. Mixtures were incubated on ice for 15–30 min, then crosslinked in a UV stratalinker 1800 (Stratagene) for two 90-sec periods with 180-degree rotation in between. The mixtures were crosslinked at a distance of 8–12 cm from the UV source to the top of an open tube. Next, 1 µl of RNAse A (2 µg/µl) or T1 (10 U/µl), or both, was added and reaction mixtures incubated at 37°C for 15 min. Equal volume of 2X SDS sample buffer was then added and reactions were boiled and then loaded onto 12% SDS PAGE gels and run at 90V for 100 min. Gels were then dried and analyzed by phosphorimager as described above.

## Immunoprecipitation of crosslinked protein-RNA complexes

For immunoprecipitations, crosslinking was performed as above. Protein A Dynabeads (Invitrogen), 10 ul per reaction, were equilibrated by washing 3X with NE buffer A (*Kataoka and Dreyfuss, 2008*), then incubated with end-over-end rotation for 1 hr at 4°C with or without 1.5 µg polyclonal rabbit hnRNP H antibody per reaction (Bethyl or ThermoFisher). Beads were then washed three more times with buffer A, resuspended in 100 µl buffer A with 0.1% Triton X-100, and mixed with crosslinking reactions, following RNAse treatment. Reaction mixtures were incubated for 30 min at 4°C with end over end rotation, then washed 3X with buffer A with 0.1% Triton X-100 adjusted to 300 mM KCl. After the third wash, isolated beads were resuspended in a 50/50 mix of buffer A and 2X SDS sample buffer (less than 20 µl total) and boiled. Tubes were placed on a magnetized stand to separate the beads, and supernatants were subjected to SDS-PAGE and bands visualized as described above.

## Purification of hnRNP H1 from *E. coli*

The cDNA sequence of hnRNP H1 was cloned in between BamHI and PstI sites of the pRSET-C vector (Invitrogen). Plasmid-containing cells were inoculated in LB medium containing 1 µg/mL ampicillin and 35 ug/mL chloramphenicol, and a 250 mL culture was grown at 200 rpm at 37°C until reaching an OD of 0.5. The culture was then induced with 1 mM IPTG for 6 hrs. Cultures were lysed in lysis buffer (1M NaCl, 50 mM $NaH_2PO_4$, 0.5 mM phenylmethyl sulphonyl fluoride (PMSF), pH 8.0) containing 0.1% Triton X-100 by sonication, then centrifuged at 8000 ×g for 15 min at 4°C. The supernatant was passed through a 0.2 µm filter, then mixed with 0.5 mL Ni-NTA Agarose beads that were pre-washed in lysis buffer, and rocked for 30 min at 4°C before being loaded onto a Poly-prep chromatography column (Bio-Rad) at 4°C. The column was washed once with 50 mL of lysis buffer, once with 25 mL lysis buffer with 8 mM imidazole, once with 20 mL lysis buffer with 40 mM imidazole, then ten times with 10 mL NaCl solution, starting with 1 M and decreasing the concentration by 100 mM each time until reaching 100 mM. The column was then eluted up to five times with 500 mM imidazole, and dialyzed against buffer D overnight. Protein was aliquoted and stored at −80°C.

## Gel shift assays

Purified H1 concentration was estimated using the A280/260 method of the NanoDrop 2000 spectrophotometer (Thermo Scientific) and diluted with binding buffer (50 mM Tris, pH 8, 50 mM NaCl, 2 mM EDTA, 0.5% BSA, 0.2% CHAPS). Reactions were mixed to a final concentration of 0.3 nM in vitro transcribed RNA, 30 nM tRNA and 0–640 nM H1 in binding buffer to a final volume of 5 ul, and incubated for 30 min at 37°C. Reactions were instantly placed on ice and 1 ul 50% glycerol was added before loading onto a pre-run native 6% acrylamide gel, and electrophoresed for 100 min at 200V. Gels were dried, imaged by phosphorimager, quantified using ImageQuant, and plotted and analyzed using Graphpad for Prism.

## Patient information and ethics statement

Human fibroblasts were isolated and grown from C9 ALS patients and normal controls under a protocol approved by the Institutional Review Board (IRB) of Columbia University. Presence of C9 repeat expansion was confirmed in Columbia University's CLIA approved molecular pathology laboratory by repeat-primed PCR. Protocols for obtaining skin biopsies and their use were approved by the IRB and all patients have signed their informed consent, including consent to publish. The procedure to culture astrocytes from post-mortem tissue had IRB approval, with the written informed consent given by the patient families.

## Skin biopsy and fibroblast culture

A skin punch kit was used to obtain biopsy samples. Once the skin had been prepared with alcohol, the area to be biopsied was anesthetized by injecting a Lidocaine solution (HCL1% and Epinephrine 1:100,000). After the skin was anesthetized, the biopsy was performed using a sterile 3 mm skin punch. A 'drilling' motion was applied by pressure and twisting until the blade of the skin punch had pierced the epidermis of the skin. Then the skin area was removed carefully into a 2 ml sterile tube with culture medium, and transferred to a laminar flow hood for culture. After the skin biopsy was rinsed with PBS, the adipose tissue was removed. Then the skin sample was cut into 6 smaller pieces, and each piece was cultured in a well from a 6 well plate using fibroblast culture medium (*Supplementary file 4*). In general, around one week, the fibroblasts come out of the skin biopsy, and were passaged into a larger flask or frozen down.

## Astrocyte culture

Human astrocyte cultures were made as described (*Re et al., 2014*). Autopsied brain tissue samples of motor cortex were collected and transferred to a laminar flow hood with a dissection microscope. Prior to cell isolation, meninges and visible blood vessels were removed, and the tissue was cut into 5 $mm^3$ cubes. The tissue was then digested for 20 min at 37°C with dissociation medium (*Supplementary file 4*), quenched with culture medium (*Supplementary file 4*), and triturated harshly with a 10 ml sterile pipette for 10 times. The solution was centrifuged at 1000 $\times$g for 3 min and the pellet was cultured into a T75 $cm^2$ flask for 2 hr in an incubator. The cell suspension was then moved to a new T75 $cm^2$ flask coated with poly-L-lysine (15 µg/ml, Sigma). Flasks were incubated in a humidified atmosphere of 5% CO2 and 95% air at 37°C for 48 hr, after which the culture medium was changed so as to remove unattached cells and myelin debris. Culture medium was changed once a week. When cultures reached confluency, the astrocytes were passaged or frozen down.

## Immunofluorescence

Fibroblasts and astrocytes (described above) were grown in their respective culture medium (*Supplementary file 4*) up to, but not exceeding, passage four. For immunofluorescence, cells were plated on Poly-L-Lysine coated 12 mm coverslips (Corning BioCoat). Cells were fixed with 3% paraformaldehyde in PBS for 15 min, permeabilized with 0.15% Triton X-100 in PBS for 15 min, and blocked with 1% BSA in PBS (filtered) with at least two PBS 5–10 min washes between each step. Prior to BSA, cells were, where indicated, treated with RNAse A or T1 by incubating coverslips at 37°C for 1 hr with either 200 ug RNAse A/mL or 1000 units RNAse T1/mL in PBS. Cells were then washed 3 times with PBS prior to incubation with 75 µl of 1.5 µg/ml BG4 for 16–18 hr at RT, then washed with PBS and blocked again with 1% BSA prior to incubation with 1ug/ml each of hnRNP H

(rabbit, Bethyl) and FLAG (mouse, Sigma) antibodies in 75 ul PBS for 3–4 hr at 37°C. One ul each of mouse-568 AlexaFluor and rabbit-488 AlexaFluor (Invitrogen) were diluted 1:1000 in 0.2% BSA in PBS, vortexed, and centrifuged for 3 min at 16,000 ×g. The upper 500 ul were then added, drop-wise, to the coverslips, which were incubated for 2.5–3 hr at 37°C. Cells were washed, stained with DAPI, and then mounted onto microscope slides, using ProLong Gold Antifade Reagent (Invitrogen). Cells were imaged using a Zeiss LSM 700 confocal microscope with a 40X objective, and pictures were taken using Zen Lite software. Each experiment used cells from patient and control cell lines, and within each experiment all cells were imaged with identical gain settings (DAPI = 690, 568 = 600, 488 = 420), and manually focused until the green (488) channel appeared at its brightest so as to minimize variation in images. The images were converted to TIF files using Image J in order to be read by the BG4 count Matlab program.

## BG4 count

BG4 Count is a script written in Matlab (MATLAB and Statistics Toolbox Release 2014b, The Math-Works, Inc., Natick, Massachusetts) that works by reading a multichannel image (given as. tif). The program first prompts the user to set a variable threshold value between 0 and 1 for the intensity of the staining to be analyzed in the red channel, which corresponds to BG4 staining. We used a threshold of 0.1 for every image. The script then removes all staining in the red channel below the threshold intensity and removes any red channel staining that does not overlap with the staining in the blue channel (nucleus, or DAPI-stained cytoplasm). It then proceeds to utilize the regionprops function to measure the parameters of each continuous object in the image and then sums the area parameter of each object in the image, AreaTotal. Then a trimmed mean of areas is taken where the top and bottom 10% of area data is removed from the data set and the remainder of the area data is averaged. This provides the average area of a single stained object. Taking the AreaTotal and dividing by this average area gives a trimmed mean approximated object count, or foci count, referred to here.

## Immunohistochemistry of spinal cord sections

Spinal cord sections were acquired from the Columbia Brain Bank and embedded in optimum cutting temperature (O.C.T.) compound (Sakura, Torrance, CA) and frozen at −80°C. Consecutive sections (25µ thick) were cut using a freezing microtome (Leica CM 3050S), mounted onto microscope slides, and stored at −80°C. Prior to fixation sections were briefly thawed and rinsed with PBS, then fixed in 4% paraformaldehyde in PBS for 15 min blocked and washed three times with PBS for 5 min each. Sections were blocked with 5% normal donkey serum diluted in Tris buffered saline (pH 7.4) with 0.2% Triton X-100 (TBS-T) and incubated with hnRNP H antibody (Bethyl) diluted in TBS-T with 5% normal donkey serum overnight at 4°C. After washing with TBS-T, tissue sections were incubated for 4 hrs at RT with species-specific secondary antibody coupled to Alexa 568 (1:1000; Life Technologies, Carlsbad, CA). After washing with TBS-T, coverslips were applied with Flouromount G (Southern Biotech, Birmingham, AL) and imaged in a blinded fashion using an SP5 Leica confocal microscope (Leica Microsystems, Wetzlar, Germany). The images were converted into z-stack max projections and individual foci were measured using ImageJ.

## siRNA transfection

U87 cells were grown as described. Cells were grown in 10 mM dishes and transfected with 25 nM si-hnRNP H (*Xiao et al., 2009*) or 25 nM negative control siRNA (Shanghai Gene Pharma) using RNAiMax (Invitrogen) as per manufacturer's instructions at 80–90% confluency. Cells were split 24 hr later, then harvested 48 hr after, and split in two fractions of 75% and 25%. The 25% fraction was subject to sonication and boiling with SDS sample buffer for western blot analysis. The 75% fraction was extracted with TriZol reagent as per the manufacturer's instructions (Invitrogen), then treated with DNAse RQI for RNA analysis.

## Western blot

Proteins were resolved on SDS-PAGE and westerns were carried out with the following antibodies: polyclonal rabbit anti-hnRNP H (Bethyl), mouse anti-Actin (Sigma), mouse mAb104. As secondary

antibodies we used HRP-conjugated anti-rabbit and anti-mouse (Sigma), and signal was detected with ECL western blotting system (GE Healthcare).

## RNA extraction from human patient cerebellum

Pulverized human brain samples were acquired from the Columbia Brain Bank. 50–70 mg of tissue was extracted on ice with 1 mL TriZol reagent by first passing through a 18 gauge needle 4–6 times, and then passing through a 22 gauge needle 3 times. The insoluble material was then pelleted at 9400 ×g for 5 min at 4°C and discarded. The TriZol layer was then mixed with 200 ul chloroform, and incubated on ice for 5 min before centrifuging at 21,000 ×g for 15 min at 4°C. The aqueous layer was removed and 0.5 volumes TriZol and an additional 200 uL chloroform were added before centrifuging again at 21,000 ×g for 15 min at 4°C. The resulting aqueous phase was mixed with equal volume chloroform, centrifuged, and the final upper phase precipitated with 1/10$^{th}$ volume 7.5 M NH$_4$OAc, 1.5 volumes cold isopropanol, and LPA by incubating at −20°C for at least 30 min, and then centrifuging at 21,000 ×g for 15 min at 4°C. The pellet was washed with 70% EtOH, air dried and resuspended in H$_2$O with 1 mM sodium citrate, pH 6.4. RNA quality was determined by electrophoresis of 1 ul of the resulting RNA on a 1% TAE agarose gel; RNA that was sufficiently intact based on the ratio of the 28S and 18S rRNA bands (approximately 2:1) was then subject to polyA selection. RNA was stored at −80°C.

## PolyA selection

20 μL of oligo(dT)$_{25}$ Magnetight (Novagen) beads were equilibrated with binding buffer (0.5 M NaCl, 10 mM Tris/HCl, pH 8) 4 times, then mixed with 18 uL undegraded RNA and 2 μL 5 M NaCl and denatured at 65°C for 10 min, followed by incubation at RT for 10 min. The supernatant was removed and the beads were washed 3 times with wash buffer (150 mM NaCl, 10 mM Tris/HCl, pH 8). The beads were then incubated with elution buffer (10 mM Tris/HCl, pH 8) for 10 min at 65°C, and separated from the beads on a magnetic stand. The elutant was precipitated with NH$_4$OAc, iso-propanol and LPA, dried, resuspended, and the concentration was measured using the NanoDrop 2000 spectrophotometer (Thermo Scientific). RNA was stored at −80°C.

## RT-PCR

Equal amounts of siRNA (negative control and hnRNP H) treated U87 RNA (100–300 ng), and equal amounts of patient cerebellum polyA RNA (50 ng) were reverse transcribed using Maxima Reverse Transcriptase according to manufacturer's instructions (Fermentas) with 4 μM random pentamer (Invitrogen). The resulting cDNAs were then diluted (1:10 for siNC/H and 1:2 for polyA cerebellum) and used in PCR with 1X standard Taq reaction buffer, 1.5 mM MgCl$_2$, 200 μM dNTPs, including 0.4 pmol α-$^{32}$P dCTP, (3000 Ci/mmol, Perkin Elmer), 0.8 μM each forward and reverse primer (*Supplementary file 3*), and 3 U/25 μl reaction recombinant Taq polymerase (Invitrogen). PCR was performed using a Biometra T Gradient thermo cycler programmed for 33 cycles of 20 s denaturation at 95°C, 30 s annealing at 55°C, and 80 s extension at 72°C, with an initial 2 min denaturation step at 95°C and a final 5 min extension at 72°C. 6X DNA load dye was added and PCR products were loaded onto a 6% native acrylamide gel, run for 2 hr at 200 V, transferred to Whatman paper, dried, and exposed to phosphorimager. Bands were quanitifed using ImageQuant, and mean percent inclusions (3–4 replicates per patient) were graphed, and statistically analyzed by t test, using Graphpad for Prism software.

## Computational analysis of RNA-seq data

Published RNA sequencing data were obtained from GSE67196 for healthy patients and c9ALS patients (*Prudencio et al.,2015*), and GSE16642 for siCtrl and sihnRNPH1 (*Xiao et al., 2009*). To identify differential AS events between healthy and c9ALS, or siCtrl and sihnRNPH1, we used rMATs (v3.0.8) (*Shen et al., 2014*) software. Statistical parameters for significant splicing change were defined as FDR <0.05 and p value<0.05. CLIPseq data of hnRNP H were obtained from GSE23694 (*Katz et al., 2010*). We mapped CLIP reads to the human genome using the software bowtie (v1.0.0) allowing up to 2 mismatches per read. After trimming CLIP short reads to 22 bp, 15487924 reads of CLIP tags were mapped uniquely to the human genome (hg38).

## Subcellular fractionation of human brain tissue

This protocol was adapted from *Jo et al., 2014*. 30–60 mg of pulverized human brain from cerebellum or motor cortex was aliquoted and weighed. For every mg of tissue, we added 18 µl of ice-cold soluble buffer (0.1 M MES (pH 7), 1 mM EDTA, 0.5 mM $MgSO_4$, 1 M sucrose) containing 50 mM N-ethylmaleimide (NEM), 1 mM NaF, 1 mM $Na_3VO_4$, 1 mM PMSF and 10 µg/ml each of aprotinin, leupeptin and pepstatin. Tissue was homogenized by 3–5 passages through a 21-gauge needle, followed by 3–5 passages through a 23-gauge needle. An equivalent volume of the homogenate was then collected from each sample and centrifuged at 50,000 ×g for 20 min at 4°C, and the remainder was stored at −80°C. The supernatant was removed and saved at −80°C as the soluble fraction, and each pellet was resuspended in 700 µl RAB buffer (100 mM MES (pH6.8), 10% sucrose, 2 mM EGTA, 0.5 mM $MgSO_4$, 500 mM NaCl, 1 mM $MgCl_2$, 10 mM $NaH_2PO_4$, 20 mM NaF) containing 1% N-lauroylsarcosine (Sarkosyl) and protease inhibitors (1 mM PMSF, 50 mM NEM and 10 µg/ml each of aprotinin, leupeptin and pepstatin), vortexed for 1 min at RT, and then incubated at 4°C overnight with end-over-end rotation. The samples were then centrifuged at 180,000 ×g for 30 min at 12°C, and the supernatant collected as the sarkosyl-soluble fraction. The pellet was resuspended in 700 µl RAB buffer and passed through a 26-gauge needle until the pellet was evenly dispersed, creating a sarkosyl insoluble fraction. Equivalent portions of sarkosyl soluble and insoluble fractions were then aliquoted and to equal volumes 2X SDS sample buffer were added to each followed by heating to 90°C. Equal volumes of each fraction were then loaded onto a 10% SDS gel and electrophoresed for 100 min at 100V followed by western blotting for hnRNP H. Densitometry was performed using ImageJ. The ratios of soluble:insoluble hnRNP H were plotted and statistically analyzed using GraphPad for Prism.

## Co-immunoprecipitation of BG4 from insoluble fraction

40 µl of the resuspended insoluble pellet prepared from motor cortex (as described in the previous section) was incubated with 100 µl IP buffer containing FLAG-M2 beads that were pre-incubated with BG4 antibody or without. Immunoprecipitation, washing and SDS PAGE were carried out as described above, followed by western blotting. Densitometry was performed using ImageJ. The amount of co-immunoprecipitatied hnRNP H was plotted and statistically analyzed using GraphPad for Prism.

## Acknowledgements

We thank Alison Maresca and Keelin Fallon for technical assistance, Tristan Coady for helpful discussions, Jessica Singleton for clinical assistance, Dr. S Balasubramanian (Cambridge) for the plasmid encoding the BG4 antibody, and the New York Brain Bank with support from Target ALS for providing us with human samples. We also thank all of the patients and their families for their generous donations. This work was supported by NIH grants R01 GM48259 and R35 GM 118136, and a grant from the Columbia Motor Neuron Center to JLM. EGC was supported in part by NIH training grant 5T32GM008798. NAS was supported by NIH grant R01 NS07377 and by a Columbia Motor Neuron Center Starter grant. LL was supported in part by the Judith and Jean Pape Adams Charitable Foundation.

## Additional information

### Competing interests

JLM: Senior editor, *eLife*. The other authors declare that no competing interests exist.

### Funding

| Funder | Grant reference number | Author |
|---|---|---|
| NIH Office of the Director | Training Grant | Erin G Conlon |
| National Institutes of Health | 5T32GM008798 | Erin G Conlon |
| Judith and Jean Pape Adams Charitable Foundation | | Lei Lu |

| National Institutes of Health | R01 NS07377 | Neil A Shneider |
| Columbia Motor Neuron Center | Starter grant | Neil A Shneider |
| National Institutes of Health | R35 GM 118136 | James L Manley |
| National Institutes of Health | R01 GM48259 | James L Manley |
| Columbia Motor Neuron Center | | James L Manley |

The funders had no role in study design, data collection and interpretation, or the decision to submit the work for publication.

### Author contributions

EGC, Conception and design, Acquisition of data, Analysis and interpretation of data, Drafting or revising the article; LL, Generated patient fibroblast and astrocyte lines, Acquisition of data; AS, Performed immunohistochemistry and imaging of spinal cord sections, Acquisition of data; TY, Performed computational analysis of RNA seq and CLIP-seq data, Analysis and interpretation of data; TT, Wrote "BG4 count", Acquisition of data; NAS, JLM, Conception and design, Analysis and interpretation of data, Drafting or revising the article

### Author ORCIDs

Aarti Sharma, http://orcid.org/0000-0002-4907-2174
James L Manley, http://orcid.org/0000-0002-8341-1459

### Ethics

Human subjects: Informed consent, including consent to publish, was obtained for human derived fibroblast and astrocyte lines used in this study by the IRB of Columbia University under protocols #AAAB0483 and #AAAC1257. For fibroblasts, written consent was given by the patients, and for astrocytes, written consent was given by the families of the deceased. Human tissue was donated for research purposes by the next of kin.

# Additional files

### Supplementary files

• Supplementary file 1. Patient and control information. CBL = cerebellum, BA4 = Brodmann Area 4, SC = spinal cord, LE = lower extremity, UE = upper extremity, MND = motor neuron disease, COPD = chronic obstructive pulmonary disease, ***minimal clinical information available.

• Supplementary file 2. Genes analyzed. Names and genomic locations of genes with cassette exon inclusion/exclusion analyzed by RT-PCR (*p<0.05, **p<0.01, ***p<0.001, ****p<0.0001).

• Supplementary file 3. Template, siRNA and primer sequences.

• Supplementary file 4. Growth medium recipes.

### Major datasets

The following previously published datasets were used:

| Author(s) | Year | Dataset title | Dataset URL | Database, license, and accessibility information |
|---|---|---|---|---|
| Prudencio M, Belzil VV, Petrucelli L | 2015 | Distinct brain transcriptome profiles in c9orf72-associated and sporadic ALS | http://www.ncbi.nlm.nih.gov/geo/query/acc.cgi?acc=GSE67196 | Publicly available at the NCBI Gene Expression Omnibus (accession no: GSE67196). |

| Xiao X, Wang Z, Jang M, Nutiu R, Wang ET, Burge CB | 2009 | Illumina mRNA-Seq of control and hnRNP H knockdown in 293T cells | http://www.ncbi.nlm.nih.gov/geo/query/acc.cgi?acc=GSE16642 | Publicly available at the NCBI Gene Expression Omnibus (accession no: GSE16642). |
| Katz Y, Wang ET, Airoldi E, Burge C | 2010 | Analysis and design of RNA sequencing experiments for identifying mRNA isoform regulation | http://www.ncbi.nlm.nih.gov/geo/query/acc.cgi?acc=GSE23694 | Publicly available at the NCBI Gene Expression Omnibus (accession no: GSE23694). |

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
