## [Decision Letter]

Thank you for submitting your article "The C9ORF72 GGGGCC expansion forms RNA G-quadruplex inclusions that sequester hnRNP H and disrupt splicing in ALS brains" for consideration by *eLife*. Your article has been favorably evaluated by Kevin Struhl as the Senior Editor and three reviewers, one of whom, Douglas L Black (Reviewer #1), is a member of our Board of Reviewing Editors.

The reviewers have discussed the reviews with one another and the Reviewing Editor has drafted this decision to help you prepare a revised submission. You will see that the required revisions are substantial and their completion may take longer than the two months we normally allow for return of a revised manuscript. The reviewers agree that these changes are necessary to bring the paper to the high level of an *eLife* paper and as a result, the revisions will be read by all three referees before a final determination can be made.

Summary:

This study from Conlon et al. examines the properties of the G_4_C_2_ repeat expansion mutation in the gene *C9ORF72*, which is the most common mutation causing the human neurodegenerative disorders Amyelotrophic Lateral Sclerosis (ALS) and Frontotemporal Dementia (FTD). The repeat has been shown to adopt a highly stable G quadruplex structure of stacked G-quartets. It is predicted that the G_4_C_2_ sequence will also bind the proteins hnRNP H and possibly F, which recognize G-runs. The authors show that a 10-repeat RNA does form quadruplexes based on its gel mobility, RNase insensitivity, and reactivity with a quadruplex-specific antibody BG4. This same RNA also crosslinks to hnRNP H, and from a variety of experiments the authors show that the protein binds to RNA that contains quadruplexes. They then examine the BG4 and hnRNP H immunostaining patterns of C9 patient and control astrocytes and fibroblasts. They find that both control and mutant cells stain with BG4 but the numbers of stained foci and their total area roughly double in the C9 mutant cells. Immunofluorescence with anti-hnRNP H antibodies identified a small number of nuclear and cytoplasmic inclusions co-stained for both BG4 and hnRNP H. These were significantly more frequent in C9 patient cells, but for most cells were not observed whether the cells were C9 mutant or wildtype. Postulating that the binding to the C9 repeat will sequester hnRNP H and reduce its activity on other transcripts, the authors compare the splicing of some known hnRNP H target exons in mutant and wildtype cells. They find modest but statistically significant changes in exon inclusion, which parallel changes seen after hnRNP H RNAi depletion. The patient cells exhibit graded splicing responses where cell lines showing the largest difference from wildtype on one exon show the largest effect for all exons. Interestingly, this gradient of hnRNP H response also parallels the severity of the disease as measured by time to death after diagnosis. Finally, the authors examine the amount of hnRNP H present in a Sarkosyl-insoluble form as a measure of the aggregated protein in cells. They find that the fraction of protein in insoluble aggregates is increased in ALS patient brain tissue over control brains and that this protein is precipitated with the BG4 anti-G quadruplex antibody.

The novelty of this paper is in the core result that the C9 repeat is binding hnRNP H. Previous work showed that C9 formed quadruplexes and one study reported possible alterations of hnRNP H target exons in mutant cells. However, ALS pathology may involve a toxic protein derived from the repeat or from other defects. Actually showing hnRNP H binding and sequestration, and clearly establishing that hnRNP H-dependent splicing was altered in ALS would be an important step understanding the pathology of the *C9ORF72* mutation. However, the reviewers were in agreement that the results on RNA binding, cellular sequestration, and splicing should all be strengthened. There are a number of issues that need to be addressed.

Essential revisions:

1) Although the data pretty clearly show hnRNP H binds to the RNA that contains quadruplexes, it is not clear how much protein is bound nor whether other proteins are also binding. While sensitive, the UV crosslinking assay is not quantitative. Is most of the RNA unbound by protein? This might be assessed in gel shift assays, by gradient fractionation, or by some kind of pull down. Can they measure an affinity? The model requires that hnRNP H be sequestered by the repeat RNA, but the cellular assays don't indicate that significant amounts are bound. Thus, it is important that at least the biochemical assays show this. Similarly, the authors report that the RNA is simultaneously forming quadruplexes and binding the protein, but the data don't distinguish protein bound to quadruplex or to other G runs in the RNA that are not in quadruplex. Results from others argue for the latter (Samatanga et al. NAR. 2013; 41(4):2505-16. PMID: 23275549). If so, then are the quadruplexes actually decreasing the amount of bound protein? Also can the authors rule out that the C9 RNA dimerizes to form the quadruplex and this gives rise to the double bands? The results also do not rule out other proteins binding to the repeats. The authors describe hnRNP H as "the major crosslinked protein", but do any other proteins crosslink at all? The intensity of crosslinking is not a necessarily a measure of the stoichiometry of binding. Given that the crosslinked protein does not IP well, it is not clear that the 70kD doublet contains only hnRNP H. This could be examined by RNA affinity purification or in analyses of the entire RNP assembled onto the C9 probe. The inability to IP the quadruplex containing RNA with hnRNP H antibodies makes it difficult to assess how much RNA is bound to the protein. Have the authors tested multiple antibodies? Do they all show this lack of binding? It would strengthen the analysis considerably if they could find an antibody that worked.

2) The described cellular phenotypes do not appear to be very penetrant. Neither the BG4 foci nor the hnRNP H aggregates are seen in more than a few cells. And the overlap of the two appears weak. Is this because of a limitation in the microscopy assays or are there no aggregates in most cells? If the latter, it doesn't support the sequestering model. They need a better assay, microscopic or other, showing that a significant fraction of the protein is bound by the repeat RNA in cells. From the data presented, it is not clear that enough protein is sequestered by the repeat to affect hnRNP H mediated splicing. It is possible that the weak overlap of the hnRNP H foci with those of BG4 is related to the inability to IP the hnRNP H when bound to the quadruplex RNA. The colocalization data might also be improved with additional hnRNP H antibodies. Alternatively one could assess the localization of Halo or GFP-tagged hnRNP H.

3) The changes in hnRNP H dependent splicing are modest. This may be sufficient to yield a cellular phenotype, but it isn't clear that they really derive from a reduction in hnRNP H activity. These exons are expected to be targeted by many proteins in addition to hnRNP H. To strengthen this correlation, they need to test many more exons. They particularly need to examine exons that do not respond to hnRNP H depletion and show that the exons changing in C9 cells are enriched in the hnRNP H targets relative to control exons.

---

## [Author Response]

*The novelty of this paper is in the core result that the C9 repeat is binding hnRNP H. Previous work showed that C9 formed quadruplexes and one study reported possible alterations of hnRNP H target exons in mutant cells. However, ALS pathology may involve a toxic protein derived from the repeat or from other defects. Actually showing hnRNP H binding and sequestration, and clearly establishing that hnRNP H-dependent splicing was altered in ALS would be an important step understanding the pathology of the C9ORF72 mutation. However, the reviewers were in agreement that the results on RNA binding, cellular sequestration, and splicing should all be strengthened. There are a number of issues that need to be addressed.*

Essential revisions:

*1) Although the data pretty clearly show hnRNP H binds to the RNA that contains quadruplexes, it is not clear how much protein is bound nor whether other proteins are also binding. While sensitive, the UV crosslinking assay is not quantitative. Is most of the RNA unbound by protein? This might be assessed in gel shift assays, by gradient fractionation, or by some kind of pull down. Can they measure an affinity? The model requires that hnRNP H be sequestered by the repeat RNA, but the cellular assays don't indicate that significant amounts are bound. Thus, it is important that at least the biochemical assays show this. Similarly, the authors report that the RNA is simultaneously forming quadruplexes and binding the protein, but the data don't distinguish protein bound to quadruplex or to other G runs in the RNA that are not in quadruplex. Results from others argue for the latter (Samatanga et al. NAR. 2013; 41(4):2505-16. PMID: 23275549). If so, then are the quadruplexes actually decreasing the amount of bound protein?*

These points together request more information on the affinity of hnRNP H for the C9 repeats and whether the interaction is sufficient to sequester significant amounts of hnRNP H. We have addressed this in several ways.

We first performed gel shift assays with purified recombinant hnRNP H to measure affinity (new Figure 2—figure supplement 1, described in subsection “HnRNP H is the major crosslinked protein”, last paragraph). The interaction between hnRNP H and C9 repeat RNA (4R, containing 4 repeats) is indeed of high affinity, supporting the physiological relevance of the proposed sequestration of hnRNP H. Specifically, we found that hnRNP H has higher affinity for a 7-deaza-GTP containing G-quadruplex (G-Q) deficient sequence (K_D_ =13.5 nM) than for the same sequence unmodified (K_D_ =75.5 nM). This suggests that G-Q formation may indeed decrease the amount of hnRNP H that is bound to C9 repeat RNA. While this aligns, as mentioned by the reviewers, with previous findings with the third qRRM (qRRM3) of hnRNP F and AGGGAU hexamers capable of forming intermolecular G-Qs (Samatanga et al., 2013), we note that our analysis used full-length hnRNP H1 and an RNA sequence that can form an intramolecular G-Q. Our measured K_D_ value with unmodified repeats (75.5 nM) was less than half that measured previously with the 7-deaza GTP modified AGGGAU hexamers and hnRNP F qRRM3 (180 nM). Based on these results, we cannot say with certainty whether hnRNP H binds G-Qs, linear G runs, or both. However, as we now state in the text: "These results suggest that hnRNP H binds either to 4R RNA in both folded and unfolded conformations but with different modes of recognition and affinities, or selectively to linear G-tracts, as has been found for the third qRRM domain of the highly related protein hnRNP F (Dominguez et al., 2010; Samatanga et al., 2013). Either mechanism, however, has relevance to the interaction of hnRNP H with the C9 repeats, as expanded transcripts are predicted to be composites of G-Qs and linear G-tracts. While determining the precise mode of binding will likely require structural studies, these results provide evidence that hnRNP H binds the C9 repeats with sufficient affinity for the interaction to be physiologically significant."

As a second means of strengthening our conclusion that a physiologically significant amount of hnRNPH is sequestered by the C9 repeats, we have improved and expanded our biochemical analysis of sarkosyl insoluble hnRN PH and hnRNP H G-Q aggregates from ALS patient and normal brains. This data is described in response to point 2 below.

*Also can the authors rule out that the C9 RNA dimerizes to form the quadruplex and this gives rise to the double bands? The results also do not rule out other proteins binding to the repeats.*

We have no evidence that the C9 RNA used in these experiments forms dimers, and are unaware of any studies suggesting this might be the case. However, while we cannot entirely rule out the possibility that the doublet reflects RNA dimerization, this would in no way affect any of our conclusions.

*The authors describe hnRNP H as "the major crosslinked protein", but do any other proteins crosslink at all? The intensity of crosslinking is not a necessarily a measure of the stoichiometry of binding. Given that the crosslinked protein does not IP well, it is not clear that the 70kD doublet contains only hnRNP H. This could be examined by RNA affinity purification or in analyses of the entire RNP assembled onto the C9 probe. The inability to IP the quadruplex containing RNA with hnRNP H antibodies makes it difficult to assess how much RNA is bound to the protein. Have the authors tested multiple antibodies? Do they all show this lack of binding? It would strengthen the analysis considerably if they could find an antibody that worked.*

Our argument that hnRNP H is the only protein to crosslink to C9 RNA was based on the fact that we detected two predominant patterns of crosslinking that were interchangeable depending on G-Q structure and efficiency of RNAse degradation. Crosslinking and immunoprecipitation (IP) provided proof that the 50 kDa species represents hnRNP H bound to short fragments of RNA. The appearance of this species was enhanced by treatments that allow degradation of unprotected RNA such as denaturing G-Qs with heat or by destabilizing G-Q tetrad formation with 7-deaza GTP. Conversely, the higher MW doublet, or 70 kDa species, represents a protein bound to full-length RNA, as proven by the RNAse insensitivity of this species. As additional evidence that the 50 and 70kDa species are related and reflect differences in RNA G-Q content, we expanded Figure 1 (new lanes 7 and 8, described in subsection “Characterization of C9 protein binding”, last paragraph) to demonstrate that incubating RNA in G-Q stabilizing conditions (100 mM KCl) during heat denaturation allows for less RNAse degradation and thus more of the 70 kDa species than when the same RNA was heated in water, which gave more of the 50 kDa species.

We agree with the reviewers that it would be preferable if we could IP the doublet with hnRNP H antibodies. We have now tried a second antibody, again without success (Figure 2, respectively). We note though that the anti G-Q BG4 antibody, which should theoretically be capable of IPing this G-Q containing complex (it does IP the C9 RNA used; Figure 1), also failed to IP the 70kDa species (data not shown). Our explanation that this species does indeed represent hnRNP H bound to the full-length RNA was therefore deductive: when the excess G-Q RNA was degraded, the only protein that remained crosslinked was hnRNP H. Thus we find it highly unlikely that hnRNP H was not bound to the full-length RNA when it contained stable G-Qs. We note that this would be true regardless of whether hnRNP H recognizes linear G-tracts, G-Qs, or both, since all of the numerous G-Q folded conformations of ten GGGGCC repeats contain some linear portion of RNA (e.g., as illustrated schematically in Figure 1). This argument is also supported by the relative sizes of hnRNP H, the free RNA, and the 70 kDa species, as well as by the gel shift data showing that both the unmodified and 7-deaza GTP modified C9 RNAs bound hnRNP H with reasonable affinities.

Despite all the above, we have now provided additional, more direct evidence that hnRNP H is the only detectable C9 crosslinking protein in U87 cell nuclear extracts (NEs). Specifically, we prepared hnRNP H-depleted NE by treating U87 cells with siRNA against hnRNP H or a negative control, and compared crosslinking using both extracts (new Figure 2, described in subsection “HnRNP H is the major crosslinked protein”, third paragraph). While we were not able to fully deplete hnRNP H, all crosslinked species, including the doublet as well as aggregated material at the top of the gel, were correspondingly reduced in depleted NE.

We believe these additional results together strengthen our conclusion that hnRNP H is the sole or at least predominant C9 repeat RNA crosslinking protein, and that the interaction is of sufficient affinity to be physiologically significant.

*2) The described cellular phenotypes do not appear to be very penetrant. Neither the BG4 foci nor the hnRNP H aggregates are seen in more than a few cells. And the overlap of the two appears weak. Is this because of a limitation in the microscopy assays or are there no aggregates in most cells? If the latter, it doesn't support the sequestering model. They need a better assay, microscopic or other, showing that a significant fraction of the protein is bound by the repeat RNA in cells. From the data presented, it is not clear that enough protein is sequestered by the repeat to affect hnRNP H mediated splicing. It is possible that the weak overlap of the hnRNP H foci with those of BG4 is related to the inability to IP the hnRNP H when bound to the quadruplex RNA. The colocalization data might also be improved with additional hnRNP H antibodies. Alternatively one could assess the localization of Halo or GFP-tagged hnRNP H.*

This point concerns the apparent paucity of hnRNP H/ G-Q foci in patient cells, and how this may detract from our conclusions. We have addressed this issue in multiple ways. First, however, we note that the statement "Neither the BG4 foci nor the hnRNP H aggregates are seen in more than a few cells" is not in fact entirely accurate: G-Q foci were detected in all cells examined, while hnRNP H foci had not been analyzed. We therefore went back and counted instances of hnRNP H foci not overlapping with BG4 in all the astrocytes and fibroblasts and found that while a majority of cells lacked detectable hnRNP H foci, they were approximately twice as common in C9ALS astrocytes and fibroblasts as in control astrocytes and fibroblasts (updated Figure 4; described in subsection “hnRNP H colocalizes with G-Q aggregates in C9 patient derived cells, but not in control cells”, third paragraph). In total, about one quarter of C9ALS astrocytes had detectable hnRNP H/BG4 and/or hnRNP H foci, while less than 5% of nonALS astrocytes had either type of hnRNP H-containing foci.

We believe, as the reviewers suggest, that the low occurrence of BG4/hnRNP H overlap detected was due to one or more technical issues. To provide evidence that excessive BG4-reactive RNA may hinder recognition of hnRNP H/G-Q foci by the hnRNP H antibody, we counted instances of hnRNP H/BG4 colocalization in the set of astrocytes in which RNAse A treatment had reduced BG4 foci to roughly equivalent numbers in C9 and nonALS cells (new Figure 4—figure supplement 1; described in the second paragraph of the aforementioned subsection). In support of this idea, we found a threefold increase in colocalization events in RNAse A-treated C9ALS astrocytes relative to untreated C9ALS astrocytes. Counting hnRNP H foci alone and hnRNP H/BG4 foci in RNAse-treated cells revealed that ~ 38% of C9ALS astrocytes show evidence of hnRNP H aggregation, compared to only 10% of control cells.

To further strengthen our microscopy data, and to establish additional disease relevance, we also imaged motor neurons (MNs) in the spinal cords of four post-mortem patients: two with C9ALS, and two controls without ALS (new Figure 4—figure supplement 2; described in the last paragraph of the aforementioned subsection). Due to high background we were unable to perform BG4 staining and instead focused on instances of hnRNP H aggregation. We counted large (>0.5 um) hnRNP H foci in the nuclei of all motor neurons identified and found 46 foci in 36 C9ALS MNs but only three such foci in 19 nonALS cells. Thus, on average, only 16% of nonALS cells contained hnRNP H foci, while the frequency in C9ALS cells was greater than one foci per cell.

While these additional analyses strengthen our conclusion that a significant fraction of cellular hnRNP H is bound by repeat RNA, we believe that our use of biochemical fractionation with sarkosyl-containing buffers (Figure 6) offers a much more biochemically accurate measure of aggregation than is possible with immunofluorescence. In order to explicitly measure insoluble, i.e. aggregated, hnRNP H, we previously presented the ratio of sarkosyl soluble: insoluble hnRNP H of only the initially insoluble fraction obtained after cell lysis. However, this analysis neglected a significant fraction of hnRNP H found in the initially soluble fraction. We therefore changed the presentation of this data to reflect the percentage of sarkosyl insoluble hnRNP H out of total cellular hnRNP H, calculated as the sum of all three fractions. While we previously included cerebellum and motor cortex from a cohort of ten individuals, we found that the overall levels of hnRNP H were more consistent in motor cortex, and thus limited our analysis to this more disease-relevant brain region. Importantly, we also added four patients to our cohort (tissue recently made available), two C9ALS, one nonALS patient, and one sporadic ALS patient with a *SOD1* mutation, confirmed negative for C9 expansion (grouped with nonALS) ([Supplementary-material SD1-data]). We conclude from this assay (new Figure 6; described in subsection “hnRNP H forms insoluble aggregates with G-Qs in C9 patient brains”, first paragraph)that on average, more than half of all hnRNP H in the cells of C9ALS patient brains was insoluble, compared to less than a third in the brains of non-C9ALS patients. This data, coupled with the related biochemical analysis of sarkosyl insoluble G-Q RNA/hnRNP H aggregates (Figure 6), provide strong support for our conclusion that sufficient amounts of hnRNP H are sequestered to bring about the splicing changes observed.

*3) The changes in hnRNP H dependent splicing are modest. This may be sufficient to yield a cellular phenotype, but it isn't clear that they really derive from a reduction in hnRNP H activity. These exons are expected to be targeted by many proteins in addition to hnRNP H. To strengthen this correlation, they need to test many more exons.*

To address these concerns, we examined splicing of an additional 11 hnRNP H target exons (for a total of 24) and 6 hnRNP A1 target exons by ^32^P-RT-PCR in post-mortem brain samples (new Figure 5—figure supplement 1–Figure 5—figure supplement 3; described in subsection “C9ALS patient brains show missplicing of hnRNP H target transcripts relative to healthy brains”). We also expanded our cohort to 14 post-mortem brain samples (7 C9ALS, 7 control), including the additional two control and two C9ALS samples mentioned above, for every gene, and also analyzed previously published CLIP-seq data for all of the exons included in our analysis (representative exons shown in new Figure 5—figure supplement 4). Briefly, this new RT-PCR data, coupled with the analyses already in the paper, strongly supports our conclusion that splicing of numerous hnRNP H target transcripts, many encoding proteins relevant to ALS/FTD, is deregulated in ALS patient brains. Additionally, examination of the CLIP-seq reads revealed that hnRNP H does indeed bind directly to all the analyzed transcripts in the vicinity of the dysregulated exons.

*They particularly need to examine exons that do not respond to hnRNP H depletion and show that the exons changing in C9 cells are enriched in the hnRNP H targets relative to control exons.*

To offer further support that the changes in exon inclusion derive specifically from hnRNP H depletion we analyzed six cassette exons that are regulated by another abundant and well-studied splicing factor, hnRNP A1, which has also been proposed as a candidate for sequestration in C9ALS (Cooper-Knock et al., 2014; Zamiri et al., 2014) (Figure 5—figure supplement 3; described in subsection “C9ALS patient brains show missplicing of hnRNP H target transcripts relative to healthy brains”, seventh paragraph). Out of these six exons, we found a significant change in inclusion between C9ALS and nonALS patients for only one, *ZCCHC17*. In support of our proposed mechanism, however, the *ZCCHC17* exon actually contains hnRNP H CLIP tags overlapping a polyG-rich region downstream of the 5’ splice site (new Figure 5—figure supplement 4), suggesting that hnRNP H promotes this exon’s inclusion. In the event of hnRNP A1 sequestration, we would expect this exon to be included more in C9ALS patients than in nonALS patients, opposite to what we observed. Only one of the other five hnRNP A1 targets (*G2AD*) had hnRNP H CLIP tags in or around the regulated exon. Similar to *ZCCHC17*, inclusion of the *G2AD* exon was also decreased slightly in C9ALS brains (p= 0.096), opposite the effect predicted for hnRNP A1 depletion. The remaining hnRNP A1 targets not found to change detectably were devoid of hnRNP H CLIP tags near the regulated exons. This data, along with analysis of three SRSF2 targets already in the paper, support our conclusion that splicing of hnRNP H targets is preferentially deregulated in ALS patient brains.